# A Comprehensive Review of mRNA Vaccines

**DOI:** 10.3390/ijms24032700

**Published:** 2023-01-31

**Authors:** Vrinda Gote, Pradeep Kumar Bolla, Nagavendra Kommineni, Arun Butreddy, Pavan Kumar Nukala, Sushesh Srivatsa Palakurthi, Wahid Khan

**Affiliations:** 1Division of Pharmacology and Pharmaceutical Sciences, School of Pharmacy, University of Missouri, 2464 Charlotte Street, Kansas City, MO 64108, USA; 2Department of Biomedical Engineering, College of Engineering, The University of Texas at El Paso, 500 W University Ave, El Paso, TX 79968, USA; 3Center for Biomedical Research, Population Council, New York, NY 10065, USA; 4Department of Pharmaceutics and Drug Delivery, School of Pharmacy, The University of Mississippi, University, MS 38677, USA; 5College of Pharmacy and Health Sciences, St. John’s University, 8000 Utopia Pkwy, Queens, NY 11439, USA; 6Department of Pharmaceutical Sciences, Irma Lerma Rangel School of Pharmacy, Texas A&M University, Kingsville, TX 78363, USA; 7Natco Research Centre, NATCO Pharma Limited, Hyderabad 500018, India

**Keywords:** mRNA structure, mRNA vaccine immune response, mRNA vaccines clinical trials, lipid nanoparticles (LNPs), cationic lipids, ionizable lipids, PEGylated lipids, lyophilized mRNA vaccines, adjuvants, antigen presentation, self-amplifying mRNA vaccines, safety, efficacy, acceptance

## Abstract

mRNA vaccines have been demonstrated as a powerful alternative to traditional conventional vaccines because of their high potency, safety and efficacy, capacity for rapid clinical development, and potential for rapid, low-cost manufacturing. These vaccines have progressed from being a mere curiosity to emerging as COVID-19 pandemic vaccine front-runners. The advancements in the field of nanotechnology for developing delivery vehicles for mRNA vaccines are highly significant. In this review we have summarized each and every aspect of the mRNA vaccine. The article describes the mRNA structure, its pharmacological function of immunity induction, lipid nanoparticles (LNPs), and the upstream, downstream, and formulation process of mRNA vaccine manufacturing. Additionally, mRNA vaccines in clinical trials are also described. A deep dive into the future perspectives of mRNA vaccines, such as its freeze-drying, delivery systems, and LNPs targeting antigen-presenting cells and dendritic cells, are also summarized.

## 1. Introduction

Vaccinations are the most effective boon for humanity for preventing the spread of infectious diseases. The impact of vaccination on the economic viability of the healthcare system is extremely large, since it lowers the treatment costs of infectious diseases. Additionally, vaccines also aid in reducing the impact and risk of outbreaks [1,2]. The wider role of vaccination in public health and safety and its extended effects on economies was reiterated and seen during the COVID-19 pandemic [2]. Successful vaccination campaigns have eradicated life-threatening infectious diseases including smallpox and polio and attempted to tackle COVID-19. The WHO estimates that vaccines prevent 2–3 million deaths each year from pertussis, tetanus, influenza, and measles [3]. Vaccines have progressed from utilizing inactivated and attenuated pathogens to subunits containing pathogen components for triggering the immune response. Important milestones in vaccine research are the development of recombinant viral-vector vaccines, virus-like particle vaccines, conjugated polysaccharide- or protein-based vaccines, and toxoid vaccines. However, the most important and a key milestone was the development of mRNA vaccines, because of its rapid development and approval for the COVID-19 pandemic and its mRNA technology producing the desired vaccine antigen intracellularly. 

We are currently in the era of mRNA vaccinations, because the groundwork research has already been laid more than three decades ago [4,5]. Although the early efforts in the 1990s to produce an effective in vitro transcribed (IVT) mRNA vaccine in animal models’ epitope presentation were effective [6,7], mRNA vaccines and therapeutics were not developed, as they were not validated until the late 1900s. Over the past decade, key technological innovations and extensive research in improving overall mRNA quality by (i) improving its stability by introducing capping, tailing, point mutations, and effective purification techniques, (ii) improving mRNA delivery by introducing lipid nanoparticles, and (iii) reducing its immunogenicity by introducing modified nucleotides, has resulted in its widespread use as a vaccine. mRNA vaccines have several important advantages as compared to the traditional vaccines including live and attenuated pathogens, subunit-based, and DNA-based vaccines. These include (i) safety, as mRNA does not integrate with the host DNA and is non-infectious; (ii) efficacy, as modifications in the mRNA structure can make the vaccine more stable and effective, with reduced immunogenicity; and (iii) manufacturing and scaleup efficiency, as mRNA vaccines are produced in a cell-free environment, hence allowing rapid, scalable, and cost-effective production. For example, a 5 L bioreactor can produce a million doses of mRNA vaccine in a single reaction [8]. Additionally, mRNA vaccines have the provision to code for multiple antigens, thus strengthening the immune response against some resilient pathogens [9]. 

The efficacy of this vaccine technology was realized when mRNA vaccines were developed and approved by Pfizer–BioNTech for the COVID-19 pandemic. These vaccines were developed in a record-breaking time of less than a year after the world was gripped with the SARS-CoV-2 virus infection, causing hospitalizations and death. This unprecedented development of Spikevax^®^ (Moderna) and Comirnaty^®^ (Pfizer–BioNTech) and their widespread vaccination to millions of people helped to control the COVID-19 outbreak. The development, approval, and manufacturing capabilities demonstrated by the makers of these vaccines has validated the mRNA platform as a safe and effective tool for vaccination. Additionally, this has also stimulated substantial interest in the scientific community to explore mRNA as a prophylactic vaccine tool. In this review, we have summarized the basics of mRNA vaccines including its mRNA structure and its pharmacological effect, mRNA structure modifications, and explained how mRNA vaccines elicit the desired immune response in the host. The review also explains the importance of lipid-based systems such as lipid nanoparticles for mRNA vaccine delivery. The article takes a deep dive into the structural components and the function of lipid nanoparticles. Recent developments in second-generation mRNA vaccines and the current clinical trials for the same are also described in detail. 

## 2. The Pharmacology of mRNA Vaccines

### 2.1. mRNA Structure

An mRNA molecule enables efficient translation of the DNA genetic sequence to the desired production of proteins by the ribosomes in the cytoplasm of the cells. Non-replicating mRNA and self-amplifying RNA are the two major types of mRNAs that are being investigated as candidate antigens for potential vaccines. The conventional, non-replicating mRNA-based vaccines encode the desired antigen for the immunogenic reaction containing the 5ʹ and 3ʹ untranslated regions (UTRs) and open reading frame (ORF), also called the coding region and the poly(A) tail. The self-amplifying mRNA contains all these components with an additional coding region in their ORF which codes for viral replication machinery which enables continuous intracellular RNA amplification followed by amplified antigen expression. In vitro transcription (IVT) is a reaction in which a linearized DNA plasmid containing the gene of interest is transcribed to the mRNA sequence. 

### 2.2. 5′ Cap

The 5′ end of the mRNA contains a 7-methylguanosine (m7G) moiety, followed by a triphosphate moiety to the first nucleotide (m7GpppN). m7GpppN is called a 5′ cap which is a protective structure which protects RNA from exonuclease cleavage, regulates pre-mRNA splicing, and initiates mRNA translation and nuclear export of the mRNA to the cytoplasm [10]. The 5′ cap is also essential in recognition of non-self mRNA or exogenous mRNA from self mRNA or the endogenous mRNA by the innate immune system [11]. The mRNA can be modified to improve its efficacy and stability by introducing many post-transcriptional modifications. Some of these include 2′-O-methylation at position 2′ of the ribose ring at the first nucleotide (Cap 1, m7GpppN1m) and the second nucleotide e (Cap 2, m7GpppN1mN2m) as well. These modifications in the 5′ cap structure not only increase the translation efficiency of mRNA but also stop activation of endosomal and cytosolic receptors, including RIG-I and MDA5, which act as defensive mechanisms against viral mRNA [11,12]. Hence, the 2′-O-methylation of the 5′ cap structure is a highly desirable property for increasing and enhancing the protein production from the mRNA after its transcription and to block any undesirable immune responses from the host immune system to the antigenic IVT mRNA. This 5′ cap can be achieved by the addition of S-adenosyl methionine and the Cap 0 structure to the IVT mRNA reaction, which yields IVT mRNA with the Cap 1 structure and S-adenosyl-L-homocysteine. Cap 1 refers to m7GpppNm, where Nm represents any nucleotide with a 2′O methylation. Trinucleotide cap analogs can also be used to make Cap 1 analogs in a co-transcriptional reaction. Ishikawa et al. utilized m7GpppAG analogs for capping IVT mRNA. These analogs in the IVT reaction permitted mRNA to have the m7G moiety at the 5′ end with no reverse-capped 5′ END mRNA products. Further modifications using nucleotides such as A, Am, m6A, or m6Am resulted in further improvised IVT mRNA specificity. Specifically, the m7Gpppm6AmG cap resulted in the maximum luciferase expression in in vitro transfection experiments in cells [13]. Sikorski et al. compared the effects of changing the first transcribed nucleotide such as A, m6A, G, C, and U with or without the 2′-O-methylationin mRNA IVT reaction. They observed that lipofectamine-delivered mRNA carrying A, Am, or m6Am as the first nucleotide resulted in a higher luciferase expression, whereas the IVT mRNA carrying G or Gm resulted in a lower luciferase expression. Importantly, the mRNA translation in a dendritic cell (DC) line JAWSII resulted in an 8-fold difference between m6A and m6Am 5′ caps. These findings prove the importance of the 5′ capping structure for efficiently targeting DCs and generating a desired immune response [14]. 

### 2.3. 5′ and 3′ UTRs

Although UTRs are not translated into the desired antigen or a protein, they are involved in regulating mRNA expression. These regions are located between the ORF and the 5′ and 3′ ends, in the upstream and the downstream of the mRNA. These UTRs contain regulatory sequences which are associated with the stability of mRNA and the efficient and correct translation of the mRNA. They also help in the recognition of mRNA by ribosomes and help in post-transcriptional modification of the mRNA [15]. The mRNA translation and its half-life can be improved by the inclusion of cis-regulatory sequences in the UTRs. Additionally, inclusion of naturally occurring sequences such as those derived from alpha- and beta-globins have been widely used to design mRNA constructs for vaccines [16,17]. Zeng et al. designed de novo 5′ UTR sequences based on the guanine–cytosine (GC) content and its (GC) length for the development of mRNA vaccines [18]. 

### 2.4. Poly(A) Tail

The IVT mRNA has a polyadenylated section at its 3′ end which is known as the poly(A) tail. This polyadenylated tail is essential for determining the lifespan of the mRNA. The poly(A) tails of the naturally occurring mRNA molecules in mammalian cells have a longer length of approximately 250 nucleotides (nt), which gets gradually shortened throughout the lifespan of mRNA in the cytosol [19]. Since the tail size affects the degradation of mRNA, the incorporation of poly(A) tails is desirable in the production of mRNA vaccines and therapeutics with longer half-life. The addition of approximately 100 nt to the poly(A) tail can result in the production of mRNA with desired prolongation of degradation [20]. 

### 2.5. Modified Nucleotides

Natural mRNA and other RNA molecules contain ATP, CTP, GTP, and UTP as the four basic nucleotides. After the post-transcriptional modification of the mRNA molecules, some of the nucleotides get modified, such as pseudouridine and 5-methylcytidine. These modified nucleotides can be utilized in the IVT transcription of the mRNA [21]. Although non-modified mRNA has its own advantages [22], modified nucleotides are beneficial in the sense that they can avoid the recognition of IVT mRNA by the innate immune system, thus avoiding any undesirable immune responses, and improving the translation efficiency of the mRNA to the desired antigen [23]. Andries et al. demonstrated that mRNAs containing the N(1)-methyl-pseudouridine (m1Ψ) modification outperformed the pseudouridine (Ψ)-modified mRNA platform by providing up to approximately 44-fold higher and 13-fold higher reporter gene expression upon transfection into cell lines or mice, respectively. The authors also demonstrated that (m5C/) m1Ψ-modified mRNA resulted in a reduction in intracellular innate immunogenicity upon in vitro transfection. The modification results in controlled activation of the toll-like receptor 3 (TLR3) and initiates the downstream innate immune signaling, which is a desired characteristic of an mRNA vaccine [24]. Figure 1 describes the structural components of an mRNA molecule. 

### 2.6. Innate and Adaptive Immune Stimulation by mRNA Vaccines 

A vaccine consisting of a pathogen-specific immunogen (encoding the viral protein) and an adjuvant can help in stimulating adaptive immune responses. While the adjuvant is designed to stimulate the innate immune response and provide the signal for T cell activation, an optimal adjuvant should stimulate innate immune response without inducing any systemic inflammation, which can elicit severe side effects. For mRNA vaccines, the mRNA molecule serves as both immunogen and adjuvant, due to the intrinsic immunostimulatory properties of mRNA. mRNA vaccination, once administered intramuscularly, leads to potential adaptive immune system activation by the following pathways: (i) transfection of muscle cells and epidermal cells, (ii) transfection of tissue-resident immune cells such as the dendritic cells (DC), macrophages, and Langerhans cells at the site of injection, thereby initiating the priming of T and B cells, and (iii) transport of secondary lymphoid tissues, such as the lymph nodes (LNs) and the spleen [25]. Figure 2 describes the modes of action of an intramuscularly administered mRNA-LNP vaccine. The host cell recognizes single-stranded RNA (ssRNA) and double-stranded RNA (dsRNA) by various endosomal and cytosolic innate receptors that form a critical part of the human innate immune response to exogenous viruses. Toll-like receptors (TLR3 and TLR7) bind to the exogenous ssRNA in the endosome, and inflammation signaling receptors including RIG-I, MDA5, NOD2, and PKR bind to ssRNA and dsRNA in the cytosol. This results in cellular activation, and generation of type I interferon and other multiple inflammatory mediators. Type I interferon has an inhibitory action on cellular translation, which can suppress the amount of the antigen produced by the mRNA vaccines. The currently available mRNA vaccines contain purified IVT mRNA which is single-stranded in nature and contains modified nucleotides. This helps in reducing the binding to TLR3 and TLR7, and immune sensors, therefore limiting excessive production of type I interferon and its inhibitory effect on cellular translation of the mRNA [26]. mRNA vaccines transfect tissue-resident immune cells, including APCs, such as DCs and macrophages [27]. 

mRNA vaccines act by transfecting the non-immune cell which leads to the production of the desired antigen. This antigen is then degraded in the proteasomes in the cytosol, which exposes the antigenic epitopes which form a complex with major histocompatibility complex (MHC) class I to the APCs such as the cytotoxic T cells expressing CD8+. This helps in establishing cellular immunity to the antigen expressed from the mRNA. Transfection of myocytes by the mRNA vaccines can activate bone-marrow-derived DCs which help in CD8+ T cell priming [28]. mRNA vaccines also act by transfecting tissue-resident immune cells, including DCs and macrophages. This triggers a local immune response at the site of injection [29]. mRNA transfection of immune cells can result in antigen presentation via MHC class I, which causes the maturation of CD8+ T cells. Additionally, the activation of the APCs can also result in presentation of the MHC class II pathway, resulting in the activation of T helper cells expressing CD4 [22]. After transfecting local immune and non-immune cells, some amount of the mRNA vaccine administered drains into the lymph nodes via the lymphatic system. The lymph nodes contain monocytes and naïve T and B cells. The transfection of the lymph node APCs can initiate priming and activation of not only T cells but also B cells [30]. Figure 3 describes the pharmacological mechanism of adaptive immune responses induced by mRNA-LNP vaccines [25]. 

## 3. Drug Delivery Technologies for mRNA Vaccines

This section may be divided by subheadings. It should provide a concise and precise description of the experimental results, their interpretation, as well as the experimental conclusions that can be drawn. mRNA vaccine molecules are large (10^4^–10^6^ Da) in size and are negatively charged. They are unable to pass through the lipid bilayer of cell membranes. Naked mRNA would be destroyed and degraded by the nucleases present in the bloodstream. In addition, naked mRNA is also attached and engulfed by immune cells in the tissue and the serum [31]. Methods to deliver mRNA molecules into the cells include techniques such as gene gun, electroporation, and ex vivo transfection. The in vivo methods of delivering mRNA involves transfection of immune or non-immune cells using lipids or transfecting agents [32]. 

### 3.1. Lipid Nanoparticles (LNPs)

Although naked mRNA, liposomes, and polyplexes have shown clinical effectiveness in humans, LNPs for mRNA vaccines are the only drug delivery system that has demonstrated clinical effectiveness and has been approved for human use. The COVID-19 mRNA vaccines against SARS-CoV-2, developed by Moderna and Pfizer/BioNTech, employ LNPs to deliver the mRNA payload to the body. LNPs are currently the foremost non-viral delivery vector employed for gene therapy [33]. The clinical effectiveness of LNPs was first demonstrated when LNP-siRNA therapeutic Onpattro^®^ (patisiran) was approved by the US FDA for hereditary transthyretin-mediated amyloidosis [34]. LNP formulations are the most successful, effective, and safe method of delivery of mRNA vaccines for human immunizations. LNPs offer numerous advantages for mRNA delivery to the site of action, including ease of formulation and scale-up, highly efficient transfection capacity, low toxicity profile, modularity, compactivity with different nucleic acid types and sizes, protection of mRNA from internal degradation, and increasing the half-life of mRNA vaccines [35]. LNPs are typically composed of four components, an ionizable cationic lipid, a helper phospholipid, cholesterol, and a PEGylated lipid. These lipids encapsulate the mRNA vaccine’s payload and protect the nucleic acid core from degradation [35]. 

### 3.2. Cationic and Ionizable Lipids

Cationic lipids were the first generation of lipids developed and utilized for mRNA vaccine delivery. These lipids contain a quaternary nitrogen atom imparting them a permanently positive charge. The positive charge of these lipids enables them to form ionic interactions with the negatively charged mRNA vaccines, forming a lipid complex called a lipoplex [4,36,37]. DOTMA and its synthetic analogue DOTAP were the first cationic lipids used to deliver mRNA vaccines in 1989 [38]. Cationic lipids such as DOTMA, DOPE, and DOGS have been widely used for mRNA delivery since then, including the commercially available and successful Lipofectin [4]. Lipofectin is a mixture of DOPE and DOTMA, and is one of the first LNP formulations, proving successful in the in vivo translation of mRNA [39]. 

The early cationic lipids demonstrated promising gene delivery in vitro, but they suffered from inadequate in vivo efficacy. The positive charge of the nitrogen head group and the non-biodegradable nature of the early cationic lipids were responsible for their ineffective delivery and efficacy in vitro [40]. Ionizable lipids, also called pH-dependent ionic lipids, are the second generation of cationic lipids containing a primary amine which imparts them a positive charge at or below physiological pH. The property of these lipids to have a neutral charge in the bloodstream at physiological pH helps in improving their safety as compared to the first-generation cationic lipids. They also extend the circulation time of the LNPs as compared to LNPs derived from cationic lipids. These were developed to overcome the shortcomings and safety issues such as immune activation and interaction with serum proteins of the first-generation cationic lipids [33]. DLin-MC3-DMA was the first US FDA-approved ionic lipid used in the first siRNA drug, Onpattro^®^ [41]. The DLin-MC3-DMA ionic lipid was synthesized after a series of modifications on the first ionic lipid DODMA. DLinDMA was formed by replacing the oleyl tails of DODMA [42,43]. DLinDMA demonstrated superior ability as compared to DODMA in protective immunity against the respiratory syncytial virus (RSV) in vivo [44]. DLinDMA was further optimized to DLin-KC2-DMA, and further to DLin-MC3-DMA depending on a series of structure–activity relationship-based studies [45,46]. DLin-MC3-DMA is considered the first generation of ionizable lipids. 

DLin-MC3-DMA or MC3 has a long plasma half-life of 72 h, increasing the duration of action of the siRNA [47]. The MC3 ionizable lipid was later shown to be effective in delivering mRNA along with siRNA [48,49,50,51,52,53,54]. The only shortcoming that MC has is its long half-life (72 h). This limits the chronic administration of vaccines with MC3. Thus, the next generation of ionizable lipids employed biodegradable functional groups which can facilitate fast clearance. The inclusion of ester moieties helped to increase the biodegradability of MC3 and increased its systemic clearance. Ester moieties are easy to install in a lipid, biodegradable, and chemically stable, which can be easily cleaved by the intracellular esterases. MC3 served as an important precursor and a starting point for the development of biodegradable ester ionizable lipids [55]. These include lipids such as Moderna’s proprietary lipids [56], Acuitas’ proprietary lipids [57], and others, including YSK12-C4 [58], CL4H6 [59], and L319 lipids, which are considered the second generation of ionizable lipids [47]. Ester-based biodegradable ionizable lipids have demonstrated higher potency in gene delivery as compared to the MC3 ionizable lipid. Moderna’s lipid 5 was found to have three-times-higher potency, and Acuitas’ lipid, ACL-0315 (the lipid used for the Pfizer/BioNTech COVID-19 vaccine), had six-times-higher potency as compared to MC3 lipid in delivering luciferase mRNA to animals US10166298B2. 

The third-generation ionizable lipids are synthesized in an optimized manner, having a limited number of chemical synthesis steps, which increases the high-throughput production of the ionizable lipids [60]. 98N12-5 is the first example of a third-generation ionizable lipid [61]. Modifications and improvements to the 98N12-5 lipidoid lead to the invention of superior analogs, including C12-200 and C14-113 [62,63]. C14-113 lipidoids can specifically target cardiac muscles and, thus, can open new vistas to optimize and target gene therapies for enhancing cardiac function [63]. Li et al. reported TT3 as a potent lipidoid for delivering various mRNA molecules encoding for CRISPR/Cas9 [64], Factor IX [65], and SARS-CoV-2 [18]. In addition to the search for enhanced efficacy, a growing interest in improving the specificity of gene delivery to specific target cells or organs is underway. Targeted delivery for vaccines and immunotherapies to the immune cells and primary and secondary lymphoid organs is rapidly underway. Some examples of targeting agents include lipids containing polycyclic tails, including 11-A-M [66], and lipids containing cyclic imidazole head groups, such as 93-O17S [66], are specifically designed to target T cells. Moreover, the cyclic amine head group in lipid A18-Iso5-2DC18 has been demonstrated to bind to the stimulator of interferon genes (STING) protein. This results in dendritic cell maturation and can have antitumor efficacy by immune stimulation [67]. This can be a useful and desired characteristic for cancer immunotherapy using gene therapy [67]. Gene therapy utilizing third-generation ionizable lipids has also shown promise for multidrug-resistant bacterial infections. Cyclic vitamin C-derived ionizable lipids delivering an anti-microbial peptide and cathepsin B mRNA to macrophages, demonstrated that the therapy can eliminate multidrug-resistant bacteria and protect the mice from bacteria-induced sepsis [68]. LNPs are the most advanced and clinically approved delivery vehicles for mRNA [69]. 

### 3.3. PEG-Lipid

Among the ingredients, polyethylene glycol (PEG) is a hydrophilic material, well known for a wide range of applications in the cosmetic, food, and pharmaceutical industries. The PEGylated lipid component in LNPs is usually linked to an anchoring lipid. PEG was found to be an essential chemical in the formulation of LNPs to mitigate the uptake of nanoparticles by filter organs, also improving the colloidal stability of LNPs in biological fluids. Hence, circulation half-life and in vivo distribution of LNPs is enhanced. Usually, PEG-lipids account for minimal molar % among lipid constituents in LNPs (approximately 1.5%). However, they play a very pivotal role in affecting crucial parameters such as population size, polydispersity index, aggregation reduction, particle stability improvement, and encapsulation efficiency. The molecular weight of PEG and the carbon chain length of the anchor lipid can be exploited to fine-tune the time of circulation and uptake by immune cells, altering the efficiency [70]. Additionally, the PEG-lipid coat on LNPs acts as a steric hydrophilic barrier for preventing self-assembly and aggregation during storage. Therefore, the presence of PEG is helpful to stabilize the LNP and regulates size by limiting the lipid fusion. The amount of PEG is inversely proportional to the size of the LNP; higher the PEG content, the smaller the size of the LNP [71]. Generally, the molecular weight of PEG ranges between 350 and 3000 Da and the carbon chain of the anchored lipid lies between 13 and 18 carbon. Multiple literature reports indicated that a higher molecular weight of PEG and longer lipid chain increases the circulation time of nanoparticles and also reduces the uptake by immune cells. As the PEG-lipid dissociates from the LNP surface, it decreases the circulation time of the LNP, and provides more chances for delivering the mRNA cargo into target cells by an effect called “PEG-Dilemma”. In some instances, as the molar% of the PEG-lipid is maintained at 1.5%.The in vivo transfection level was found to be independent of the carbon chain length of the lipid. An added advantage of PEG-lipids relies on their capability of conjugating a specific ligand to the LNP, thus aiding in targeted drug delivery [72,73]. 

### 3.4. Helper Lipids

The main function of helper lipids in the formulation of LNPs lies in supporting their stability during storage and in vivo circulation. Chemically, these are glycerolipids and non-cationic in nature. Among the various helper lipids, sterols and phospholipids are the most widely used. Cholesterol is a natural component present in cell membranes. It is an exchangeable moiety that can be easily accumulated in the LNP. From a series of different studies, it has been indicated that cholesterol might be present on the surface, within the lipid bilayer, or even conjugated with the ionized lipid within its core. It is usually incorporated in LNP formulation, to maintain stability by filling gaps between lipids. The presence of cholesterol is needed to regulate the density, uptake, and fluidity of the lipid bilayer matrix within the LNP. Therefore, it controls the rigidity and integrity of the membrane, thereby preventing any leaks by the “condensing effect”. The hydrophobic tail, sterol ring flexibility, and polarity of hydroxy groups in cholesterol was reported to impact the efficacy of LNP delivery [74]. Cholesterol also contributes to improving the circulation half-life of LNPs by reducing the surface-bound protein. Moreover, it helps by fusing with the endosomal membrane during the cellular uptake of LNPs. It plays a vital role in lowering the temperature needed for transitioning from the lamellar phase to the hexagonal phase; therefore, the mRNA cargo from the LNP will be delivered to the cytosol [75].

The inclusion of phospholipids in LNP formulation can help with boosting encapsulation (together with cholesterol) and increasing cellular delivery. In general, the number of phospholipids in the LNP is considerably reduced, while increasing the cholesterol content for longer circulation times. Additionally, the inclusion of phospholipids promotes the entrapment efficiency and transfection potency of the LNP. It has been reported that increasing the molar percentage of phospholipids contributes to expediting the efficacy of delivery by LNPs. These phospholipids in Zwitter ionic form have been reported to play a pivotal role in the assembly of the LNP through the stabilization of electrostatic interactions between the cationic lipid, mRNA cargo, and surrounding water molecules. However, the actual role of phospholipids in the delivery of mRNA via LNPs is still ambiguous. Hence, it remains intriguing to further explore the actual role of phospholipids in enhancing the particle stability and delivery in vivo. Figure 4 describes the components of LNPs, including ionizable lipids, cholesterol, helper lipids, and PEGylated lipids [35].

### 3.5. Physicochemical Properties Affecting mRNA-LNPs

LNPs possess many distinct characteristics, with a majority of being beneficial; ironically, a few characteristics grant some unwanted toxicities. Therefore, it is very critical to understand the physicochemical properties that affect the mRNA-loaded LNP.

**Size and Surface area:** Size and surface area dictates the LNP interaction process with the biological system, along with the distribution, elimination, internalization, degradation, and response. Decreasing the size corresponds to an increase in the surface area, thus making it more reactive towards the surrounding biological milieu. Essential biological activities including endocytosis and cellular uptake rely mostly on the particle size. Any size-dependent toxicity is based on the ability of LNPs in entering the biological system and modifying the macromolecules, thereby altering the essential biological functions. In the case of vaccines, high and efficient delivery is reported, while maintaining a particle size of ≅50 nm, irrespective of its chemical composition. 

**Charge:** Charge plays a primary role in deciding the fate of biodistribution and efficacy of LNPs. The charge of the vector is very instrumental in transporting mRNA vaccines across biological membranes. Hence, negatively charged mRNA can develop electrostatic interactions with positively charged cationic lipids, leading to efficient encapsulation. In the end, the cationic liposome interacts with the anionic cell surface and endosomal membrane to release the mRNA cargo. The pKa (ability to attain positive charge) of cationic lipids has a significant effect on delivering the mRNA cargo; apparently, it is very important to understand its role. Although, there remains some uncertainty surrounding the actual pKa needed for gene delivery. A few reports indicated that the ideal pKa range for the delivery of LNPs via the IV route is in between 6.2 and 6.6. Charge modulation has effectively been researched for mitigating toxic manifestations, along with improving the delivery of mRNA from LNPs.

**Shape and Structure:** Both shape and internal structure are essential parameters that directly influence the cellular uptake and interaction with the biological environment. A few reports mentioned that the endocytosis of spherical nanoparticles is relatively easier in comparison to other shapes. Alternatively, non-spherical nanoparticles are more inclined to flow through capillaries. The exact mechanism of action underlying the shape and structure and its role in vivo remains obscure to date. Due to the involvement of many technological challenges, the actual mechanism of action stemming from shape and structure remains widely unexplored. Therefore, the research needs to be accelerated towards understanding their activity in deforming membranes and therapeutic efficiency. 

**Surface Composition:** Efficient delivery and the biodistribution of LNPs can be influenced by the surface composition of delivering vectors. Well-known examples include the surface modification of LNPs by incorporating the PEG-Lipids by PEGylation. This process of PEGylation is known to alter nanocarrier trafficking and extend circulation half-life. Nevertheless, along with improving biodistribution and circulation, PEGylation can also result in reducing the uptake of LNPs by steric hinderance and limits interactions with the plasma membrane. Hence, the PEG-lipids detach into the serum and alleviate the steric hinderance to favor endosomal uptake [72,76].

## 4. mRNA Vaccines Manufacturing 

mRNA vaccines have demonstrated several advantages over traditional vaccines, including the ease of their development, easy scale-up, and rapid manufacturing. Similar to other vaccines, mRNA vaccine drug products undergo three typical steps in their manufacturing, which are upstream production, downstream purification, and finally formulation of the mRNA drug substance. This section will discuss these steps and newer developments in each process to streamline mRNA vaccine production. 

### 4.1. Upstream Production

The upstream production of mRNA vaccines comprises the generation of the mRNA transcript from the plasmid containing the gene of interest. This reaction is called the in vitro transcription reaction (IVT). The IVT enzymatic reaction relies on RNA polymerase enzymes such as T7, SP6, or T3. The RNA polymerase enzymes catalyze the synthesis of the target mRNA from the linearized DNA template containing the gene of interest. A linearized DNA template is produced by the cleavage of a plasmid containing the gene of interest by restriction of endonucleases enzymes, or alternatively, amplification of the gene of interest by PCR can also produce mRNA molecules. The essential enzymes of an IVT reaction include: (i) RNA polymerase—converts DNA to RNA, (ii) inorganic pyrophosphatase (IPP)—increases IVT reaction yield, (iii) guanylyl transferase—adds GMP nucleoside to 5′ end of mRNA, (iv) Cap 2′-O-Methyltransferase (SAM)—this enzyme adds a methyl group at the 2′ position of the 5′ cap of the mRNA, (v) DNase I—endonuclease used for removal of contaminating genomic DNA from RNA samples and degradation of DNA templates in the IVT reaction, and (vi) poly(A) tail polymerase and (vii) modified and unmodified nucleoside triphosphates (NTPs). These enzymes facilitate the upstream development of the mRNA transcript from a plasmid containing the gene of interest. The capping enzymes include SAM and guanylyl transferase which enzymatically form a 5′ cap at the 5′ end of the mRNA, while the poly(A) tail polymerase tailing enzyme forms the poly(A) tail. Another method of 5′ capping is using the co-transcriptional method, where the 5′ cap is prepared previously, and this cap is added to the mRNA in a non-enzymatic manner. This co-transcription reaction can be performed using CleanCap^®^ Reagent AG [77]. 

### 4.2. Downstream Purification

mRNA is produced by the IVT reaction in the upstream production phase; it is then isolated and purified by multiple purification steps in downstream processing. The IVT reaction mixture contains several impurities including residual NTPs, enzymes, incorrectly formed mRNAs, and DNA plasmid templates. Lab-scale purification of IVT mRNA involves methods based on DNA removal by DNase enzyme digestion followed by lithium chloride (LiCl) precipitation [78]. The lab-based methods do not allow the complete removal of aberrant mRNA species including dsRNA and truncated RNA fragments. The removal of these impurities is essential and critical to obtain a pure mRNA product which demonstrates its intended efficacy and safety profile. An inefficient purification technique can result in the mRNA vaccine product having decreased translation efficiency and an unwanted immunostimulatory profile. For example, a 10–1000-fold increase in mRNA transfection and related protein production was observed when modified mRNA was purified by reverse-phase HPLC prior to its delivery to dendritic cells [79]. 

Chromatography is a commonly and widely used purification process accepted in the biopharmaceutical industry for the purification of vaccines and biologic drug products. The first published procedure in 2004 for large-scale nucleic acid purification of RNA oligonucleotides used size exclusion chromatography (SEC) [80,81]. SEC has several advantages including selectivity, scalability, versatility, cost-effectiveness, and achieving high purity and yields for nucleic acid products. However, SEC cannot remove impurities having the same size, such as dsDNA. Instead of SEC, ion-pair reverse-phase chromatography (IEC) has demonstrated to be an excellent purification technique for mRNA vaccines [79,82,83]. IEC can easily separate the target mRNA from the IVT reaction impurities. This separation method relies on the charge difference between the target mRNA and the impurities. IEC has several advantages including separation of longer RNA transcripts from the target mRNA, higher binding capacity, cost effectiveness, and scalability. Since IEC is performed under denaturing conditions, the process becomes complex and temperature-sensitive [84]. Affinity-based chromatographic separation is another mRNA purification method. Deoxythymidine (dT)-Oligo dT is a sequence that captures the poly(A) tail of the mRNA. Chromatographic beads containing Oligo dT can be used for the downstream purification of mRNA vaccines [85]. Tangential flow filtration (TFF) or core bead filtration can be utilized for the removal of small-sized impurities [86]. As a final polishing step for mRNA vaccines, hydrophobic interaction chromatography (HIC) connected to a connective interaction media monolith (CIM) column containing OH or SO3 ligands can be extremely beneficial [86]. 

### 4.3. Formulation 

mRNA molecules, being negatively charged, should be formulated in a lipid-based drug delivery system for avoiding mRNA degradation and improving its transfection efficiency and half-life. LNPs are the most trustworthy, reliable, and US FDA-approved lipid-based non-viral carrier system for delivering mRNA vaccine drug substances. mRNA LNPs are formed by precipitating lipids dissolved in an organic phase and mixing them with mRNA in an aqueous phase. The most commonly used lipids in the organic phase are ionizable lipids, cholesterol, helper lipids, and PEG-lipids. Meanwhile, the mRNA is dissolved in a citrate or acetate buffer at pH 4. Mixing the aqueous and non-aqueous solutions protonates the ionizable lipid, causing electrostatic attraction between the ionizable protonated lipid and the anionic mRNA. This interaction is simultaneously coupled with the hydrophobic interactions of other lipids and drives a spontaneous self-assembly of the mRNA-LNPs with the mRNA encapsulated within the core of the nanoparticles. This process is also called microprecipitation. Following LNP formation, they are dialyzed to remove the non-aqueous solvent, which is usually ethanol, and elevate the solution pH to physiological pH. Microfluidic mixers enable the formation of small-sized LNPs with a low polydispersity index and high mRNA encapsulation efficiency. Microfluidic mixing is the most commonly used method for mRNA LNP formulation at the lab scale and for GMP level as well. Precision NanoSystems’ NanoAssemblr^®^ platform has been widely utilized for LNP formulation development and GMP production under controlled environments [87]. This system uses a staggered herringbone micromixer (SHM) cartridge architecture. The structure of SHMs enables the two aqueous and non-aqueous solvents to mix within microseconds. This timescale is much smaller than the time required for lipid aggregation; hence, SHMs produce small nanoparticles of uniform size [87]. The NanoAssemblr^®^ settings can be simply adjusted to change the flow rate and volume of the aqueous and the non-aqueous phase to obtain LNPs of the desired size and size distribution. A total flow rate of 12–14 mL/min and a flow rate volume ratio of 3:1, non-aqueous:aqueous phase, is commonly used to generate small monodisperse LNPs. Although SHMs have several advantages for efficient production of LNPs, their utility GMP manufacturing is limited due to solvent incompatibility. The long-term exposure of the SMH and its internal parts containing polydimethylsiloxane to ethanol can lead to its deformation. It becomes difficult to replace the cartridges in a continuous GMP manufacturing run. Hence T-mixers are utilized for LNP scale-up and manufacturing. They can produce LNPs similar to the SMH, can handle higher flow rates and volumes (60–80 mL/min), and are compatible with organic solvents such as ethanol [87,88,89]. Figure 5 explains the processes of mRNA vaccines manufacturing [86]. 

## 5. mRNA Vaccines in Clinical Trials 

The critical step for any vaccine candidate after successful preclinical studies and prior to market launch is clinical development. The clinical development of any mRNA vaccine consists of a series of clinical trials to evaluate the safety, immunogenicity, and efficacy in humans. Based on the patient population and objectives of the trial, they are categorized as Phase 1, 2, 3, and 4. Phase 1 studies are conducted in a small group of humans (ideally one center) mainly to determine the safety and pharmacokinetics of the vaccine. Phase 2 studies are proof-of-concept studies mainly intended to confirm the results obtained in Phase 1 studies and evaluate the efficacy in a slightly higher number of humans [91,92]. Phase 3 studies are confirmatory studies conducted in multiple centers and in a wide range of the human population to confirm the efficacy and safety of the vaccine candidate. These studies are usually conducted with an active comparator or placebo. Phase 4 studies are conducted after the market approval of the vaccine candidate and are mainly aimed at confirming the safety of the vaccine. Each vaccine candidate should undergo critical clinical evaluation in all the clinical studies before its commercial launch. The development of a vaccine takes a few years to complete. However, Comirnaty (Pfizer) and Spikevax (Moderna) obtained Emergency Use Authorization (EUA) in less than a year due to the COVID-19 pandemic. Currently, there are a wide variety of mRNA vaccines in clinical trials intended for infectious diseases (COVID-19, influenza, Zika virus, Nipah virus, respiratory syncytial virus, and others), genetic disorders, and cancers due to the ability of the mRNA vaccine to balance both adaptive as well as innate immune responses. Majority of these vaccines are liposome-based and are in Phase 1 and 2 clinical trials. Moreover, around 60–70% of the ongoing clinical studies are being conducted with mRNA-based COVID-19 vaccines. Therefore, we have summarized all the ongoing clinical trials with mRNA-based vaccines (excluding COVID-19 vaccines) in Table 1 below. As mentioned earlier, the majority of the mRNA vaccines are in the early phases of clinical trials (Phase 1 or 2) and only a few mRNA-based vaccines are in Phase 3 development. The sections below provide a detailed information about the vaccines currently in the Phase 3 stage [5,93,94]. 

### 5.1. mRNA-1345

mRNA-1345 is a vaccine candidate developed by Moderna for respiratory syncytial virus (RSV) infection, which encodes for an RSV protein known as prefusion F glycoprotein, thus eliciting an efficient neutralizing antibody response. This protein is responsible for the entry of the virus and cell-to-cell spread and is critical in the propagation of RSV infection. This vaccine is a lipid nanoparticle-based vaccine consisting of optimized protein and codon sequences. US FDA has recently granted a fast-track review designation for mRNA-1345 for adults > 60 years of age. Several vaccines prior to mRNA-1345 developed for RSV infection have failed in clinical trials due to low immune response [95]. Recently, Moderna has reported interim results of the ongoing Phase 1 study evaluating tolerability, reactogenicity, and immunogenicity of mRNA-1345 in children, younger adults, older adults, and women of child-bearing age. Results showed that the vaccine was well tolerated at all the dose levels in the trial as of the data cut-off date. The study is expected to be completed in 2023. A Phase 2/3 study of mRNA-1345 vaccine (NCT05127434) in adults ≥ 60 years of age is being conducted to evaluate the safety and tolerability of the mRNA-1345 vaccine and to demonstrate the efficacy of a single dose of the mRNA-1345 vaccine in the prevention of a first episode of RSV-associated lower respiratory tract disease (RSV-LRTD) as compared with placebo from 14 days post-injection through 12 months. The study is planned to be conducted in two placebo-controlled phases, i.e., Phase 2 in 400 to 2000 participants and Phase 3 in >30,000 participants. The primary objective of the study is to evaluate the safety and efficacy of the vaccine. Safety endpoints include the monitoring of participants for the incidence of adverse reactions, adverse events, serious adverse events, and adverse events of special interest. The primary efficacy endpoint includes the Vaccine Efficacy (VE) of mRNA-1345 to Prevent a First Episode of RSV-LRTD within the period of 14 Days post-injection up to 12 Months post-injection. This study was started in November 2021 and is expected to be completed by November 2024 NCT05127434. 

### 5.2. mRNA-1010

mRNA-1010 is a quadrivalent vaccine candidate developed by Moderna for flu, which encodes for the surface protein, hemagglutinin (HA) protein from four seasonal influenza viruses based on the recommendations of the World Health Organization, including seasonal influenza A/H1N1, A/H3N2, and influenza B/Yamagata-, and B/Victoria-lineages. HA is considered as an important target for vaccine development as it generates broad protection against influenza and is the primary target of currently available influenza vaccines. The efficacy of mRNA-1010 has been evaluated in Phase 1 and Phase 2 studies. In December 2021, Moderna released interim results of the ongoing Phase 1 study which evaluated mRNA-1010 at 3 doses (50 µg, 100 µg, and 200 µg) in younger and older adults. Results showed that RNA-1010 successfully boosted hemagglutination inhibition assay geometric mean titers against all strains 29 days after vaccination at all doses in all the participants with no significant safety findings. The company also confirmed that the ongoing Phase 2 study with mRNA-1010 has reached the full enrollment and an interim analysis is planned in 2022. A Phase 3 active-controlled study (NCT05415462) is being conducted to evaluate the immunogenicity and safety of mRNA-1010 seasonal influenza vaccine in adults ≥ 18 years. The active comparator is any licensed quadrivalent inactivated seasonal influenza vaccine. The primary objectives of this study are to evaluate the humoral immunogenicity of mRNA-1010 relative to that of an active comparator against vaccine-matched influenza A and B strains at Day 29, and to evaluate the safety and reactogenicity of mRNA-1010. Safety endpoints include the monitoring of participants for the incidence of adverse reactions, adverse events, serious adverse events, and adverse events of special interest. Primary efficacy endpoints include geometric mean titer (GMT) of anti-hemagglutinin (HA) antibodies at day 29 and percentage of participants reaching seroconversion. This study was started in June 2022 and is expected to be completed by August 2023 NCT05415462. 

### 5.3. mRNA-1647

mRNA-1647 is a vaccine candidate developed by Moderna for cytomegalovirus (CMV) infection in women of childbearing age. It consists of six mRNAs which encodes for two antigens on the surface of CMV. Five mRNAs encode the subunits that form the membrane-bound pentamer complex, while the sixth encodes the full-length membrane-bound glycoprotein B (gB). The mRNA-1647 vaccine instructs human cells to manufacture the antigens, resulting in functional antigens that mimic those presented to the immune system by CMV during a natural infection. To date, the mRNA-1647 vaccine has been evaluated in Phase 1 and Phase 2 studies. Interim analysis results of these two studies were positive and led to the start of a Phase 3 study to confirm the efficacy and safety of mRNA-1647. This Phase 3 study (NCT05085366) is a randomized, observer-blind, placebo-controlled study to evaluate the efficacy, safety, and immunogenicity of the mRNA-1647 vaccine in healthy participants 16 to 40 years of age. The primary objective of the study is to evaluate the efficacy of the mRNA 1647 vaccine in CMV-seronegative female participants and to evaluate the safety and reactogenicity of the mRNA-1647 vaccine in all participants. Safety endpoints include the monitoring of participants for the incidence of adverse reactions, adverse events, serious adverse events, and adverse events of special interest. Primary efficacy endpoints include seroconversion from a negative to a positive result for serum immunoglobulin g (IgG) against antigens not encoded by mRNA-1647 (Time Frame: Day 197 (28 days after the third injection) up to Day 887 (24 months after the third injection)). The study was started in October 2021 and expected to be completed by July 2025 NCT05085366. 

### 5.4. Clinical Safety of mRNA-Based Vaccines

The main purpose of early-stage clinical trials in the clinical development program of any vaccine candidate is to evaluate its safety in a human population. The safety of the vaccine is evaluated throughout the clinical development by mainly monitoring adverse events, deaths, laboratory findings, and others. The marketing authorization/approval of any vaccine is possible only if the safety profile is acceptable. Regulatory agencies/Institutional Ethics Committees (IEC) can stop the clinical trial if there are any untoward events during the study and the clinical development program can be halted. It is expected that the adverse events associated with the vaccine candidate should be resolved/recovered quickly [5]. Even after the marketing approval of the vaccines, the sponsors are responsible for monitoring the safety profile. Toxicity is one of the important factors that needs to be considered for the mRNA-based vaccines due to the presence of nucleosides. It is reported in the literature that toxicity of some nucleoside-based anti-cancer drugs and antivirals drugs is due to unnatural nucleosides [94]. Specific to mRNA vaccines, hepatotoxicity was the most common toxicity observed during the preclinical studies for a vaccine in development for Crigler–Najjar syndrome. This could be attributed to the presence of any toxic excipient during the formulation of the lipid nanoparticles used for delivery. In another study with mRNA vaccine for rabies, systemic adverse events were reported in a clinical trial due to the inflammatory nature of mRNA. The majority of the toxicities associated with mRNA vaccines are mainly due to the excipients used for the formulation or other solvents used during the formulation development. These toxicities can be avoided by using excipients within their safety limits and by following processes that reduce the residual toxic components in the vaccine. In the future, the expected toxicities with mRNA-based vaccines include local and systemic inflammation, the biodistribution and persistence of the expressed immunogen, stimulation of auto-reactive antibodies, and potential toxic effects of any non-native nucleotides and delivery system components. In addition to the above, these vaccines could also induce potent type I interferon responses, edema (due to extracellular naked RNA), blood coagulation, and pathological thrombus formation [96]. On the other hand, several mRNA vaccines have been approved for human use by the global health authorities. All the approved vaccines have shown acceptable safety profiles during their evaluation in clinical trials. For example, the two COVID-19 mRNA vaccines from Pfizer–BioNTech (Comirnaty) and Moderna (Spikevax) have demonstrated excellent safety and efficacy profiles. Overall, the safety profiles of several mRNA-based vaccines in clinical development have been acceptable (well tolerated) and very few to none have been withdrawn from clinical trials to date. The majority of the adverse events reported during the clinical studies include injection site reactions. All the sponsors are expected to consider safety as an important factor during vaccine development by conducting thorough toxicity testing during nonclinical development. The observations from the nonclinical studies should be taken into consideration during the clinical trials and should be monitored carefully [97,98].

## 6. Secondgeneration mRNA Vaccines

The second-generation vaccines are the ones developed after improving some of the inefficiencies and enhancing the safety, efficacy, storage, and handling of the former humble first-generation mRNA vaccines. The changes involve making the vaccines stable at room temperatures and reduce the requirement of a cold chain for their storage and transportation, while maintaining the same efficacy and safety. Other changes involve finding more potent and ligand-targeted nanocarriers which can have a better safety and mRNA delivery efficacy profile. Additionally, immense research about exploring various RNA-based molecules for use as vaccines including self-amplifying RNA is ongoing. This section highlights the second generation of mRNA vaccines which can be seen developing in the near future. 

### 6.1. Lyophilized mRNA Lipid Nanoparticles

Typically, mRNA lipid nanoparticle (LPN) vaccines must be stored at a subzero temperature to maintain stability and efficacy. The use of cold-chain shipping for the storage of vaccines limits the access of vaccines in low- and emerging-economy countries [99]. Long-term stability is one major concern for the development of LNPs owing to the physical and chemical instabilities observed when LNPs are stored as an aqueous suspension [100]. Chemical degradation involves alteration of bonds in the mRNA molecule [101]. Physical degradation encompasses the denaturation/aggregation (loss of secondary and tertiary structure), fusion, and leakage of encapsulated mRNA. Chemical degradation of mRNA predominantly occurs via hydrolysis and oxidation [102]. Degradation of lipids in LPNs also leads to hydrolysis and oxidation. Hydrolysis mainly occurs through the phosphodiester bonds, a backbone of the mRNA molecule. Oxidation, on the other hand, affects the nucleobases and sugar groups of mRNAs. Oxidation results in mRNA strand break, cleavage of bases, and change in secondary structure, which can stop translation of antigen in vivo. Furthermore, lipid crystallization and lipid polymorphic transformations during storage of LNPs result in leakage or drug expulsion [103]. Storage conditions are one critical parameter that strongly influences the stability of mRNA-LNP vaccines. The long-term storage of mRNA-LNPs is not yet fully explored. During prolonged storage, mRNA-LNPs may undergo structural changes. Hence, it is crucial to know what changes are imposed on it in order to obtain a stable mRNA-LNP during storage. As the degradation reactions in mRNA-LNPs are initiated by the presence of water, lyophilization is one widely used drying technique for the long-term storage of nanoparticles including mRNA-LNPs. In lyophilized cake or powder form, the mRNA-LNP vaccines can be conveniently shipped worldwide without requiring freezing storage. 

The lyophilization process is divided into three stages: freezing, primary drying, and secondary drying [104]. During the freezing process, water is frozen into ice crystals while solute materials are excluded as the cryo-concentrated phase. Both primary and secondary drying processes are carried out under a vacuum. The primary drying step is carried out at low temperatures, and frozen water is removed through the sublimation process during this phase. The temperature is increased during the secondary drying step to remove any unfrozen water by a desorption mechanism. The details of the lyophilization process are discussed in other studies [104,105,106]. 

As the mRNA-LPN vaccines are prepared from specific lipid types at a certain concentration, it is important to retain physicochemical parameters such as particle size, polydispersity, and encapsulation efficiency during lyophilization and subsequent storage. Thus, careful selection of lyophilization process parameters, buffers, and cryo- and lyoprotectants is of utmost importance to ensure the stabilization effect. In a study, Shirane et al. [107] lyophilized dispersions of ethanol containing siRNA-LNP after the formation of LNPs. The results show that there is no difference in in vivo gene knockdown efficiency between freshly prepared (conventional) and reconstituted lyophilized formulations, demonstrating the feasibility of lyophilization of mRNA-LNPs. The mRNA entrapped in the LNP may be exposed to different stresses during the freezing and drying steps of the lyophilization process, which ultimately affects the stability of the mRNA-LNP. Hence, it is important to use cryoprotectants in the formulation, and the optimal cryoprotectant for the stabilization of LNPs depends on the material and type of formulation [108].

Zhao et al. [109] identified optimal storage conditions for lipid-like nanoparticles (LLNs) of mRNA. The LLNs were prepared using ionizable lipid, N1, N3, N5-tris(3-(didodecylamino) propyl) benzene-1,3,5-tricarboxamide derivative, TT3. The types of cryoprotectants (trehalose, glucose, and mannitol) and physical state conditions such as aqueous, freezing, or lyophilized were screened and evaluated for properties such as nanoparticle size and mRNA expression in vitro and in vivo. In aqueous conditions, LLN-mRNA did not maintain long-term storage stability. The addition of cryoprotectants at an optimal concentration helps in retaining the in vitro expression efficiency of mRNA in lyophilized LLNs. However, during the lyophilization and reconstitution process, the nanostructure of LLN-mRNA is altered, affecting the in vivo interaction of mRNA-LLNs with serum proteins, leading to different in vivo efficiency. Freezing LLNs of mRNA in liquid nitrogen with 5% sucrose or trehalose was found to be optimal for long-term storage.

Hong et al. [110] developed a lyophilizable SARS-CoV-2 vaccine using a cationic lipid-based delivery system. The reconstituted vaccine induces the humoral and cellular immune response to SARS-CoV-2 in mice, demonstrating the immunogenicity and neutralizing antibody activity of SARS-CoV-2 after lyophilization. 

A variety of stresses during freezing could impact the stability of LPNs, including crystal formation, interfacial effects, freeze-concentration, buffer pH change, and phase separation [111]. Crystal formation during freezing imposes an ice–liquid interface, which can lead to adsorption and damage of the colloidal structure of protein molecules, including mRNA. Freezing increases the concentration of solute material in the remaining liquid fraction, facilitates particle–particle interactions, and results in particle aggregation [112]. Freeze concentration increases the osmotic pressure on the lipid bilayer, imparts physical stress to the membrane, and causes membrane rupture. Furthermore, osmotic stability depends on the lipid membrane composition owing to selective solute permeability. The details on freezing- and drying-induced stresses are discussed in other studies [113,114,115]. 

In one study, Jones et al. [116] examined the effect of freeze-drying on the integrity of mRNA. The purified RNA at 25 µg/mL was freeze-dried in water or 10% trehalose, stored at −70°, −20°, 4°, 37 °C, or room temperature under nitrogen gas for up to 10 months and analyzed for RNA integrity. The recovery of freeze-dried RNA in water varied between 66% to zero of that of freeze-dried RNA in 10% trehalose. Furthermore, RNA stored in 10% trehalose showed high consistent recovery for all time points, permitting the storage of RNA at 4 °C for up to 10 months. This allows RNA vaccine development even in developing countries.

Muramatsu et al. [117] demonstrated that nucleoside-modified mRNA-LNPs can be lyophilized and that the physicochemical properties (particle size, encapsulation efficiency) of the mRNA LNPs did not change significantly after 12 weeks at ambient temperature and at least 24 weeks at 4 °C storage. However, a 10–15% and 30% decrease in RNA integrity was observed for lyophilized nucleoside-modified mRNA-LNPs when stored at 4 °C and 25 °C, respectively. Furthermore, in vivo bioluminescence imaging studies in mice show that the lyophilized firefly luciferase-encoding mRNA-LNPs retain their high expression without losing their high translatability. In the comparative mouse immunization studies, the authors demonstrated that the potency of the lyophilized nucleoside-modified mRNA LNP influenza virus vaccine was retained after 12 weeks of room-temperature storage or for at least 24 weeks after storage at 4 °C. 

Water replacement hypothesis and devitrification are two mechanisms by which cryo- and lyoprotectants (sugars) stabilize biological systems during lyophilization [104]. Trehalose as a lyoprotectant is reported to stabilize the mRNA–protamine complex formulations both during the freeze-drying process and subsequent storage at −80, 5, 25, and 40 °C. The quality attributes analyzed during the storage period, including appearance, RNA integrity, RNA content, pH value, and osmolality, have met the stability specifications required for a quality (stable and safe) RNA medicament (WO2016165831A1).

Sucrose also appears to be a suitable cryoprotectant for stabilization of SS-cleavable proton-activated lipid-like material (ssPalm), a component of LNPs. Typically, during freezing, the molecules with less mutual miscibility lead to phase separation, and the particles will undergo aggregation/coalescence via mutual collision. Sucrose possesses high miscibility with an LNP surface containing a grafted polyethylene glycol (PEG) polymer. The preferential interaction between the sucrose and PEG layer on the LNP surface stabilizes the particles by exhibiting cryoprotective properties [107]. 

The interaction between the sugars and the phospholipid head group reduces the lipid membrane melting temperature in the dry state. Sugars reduce the van der Waals interactions among the acyl chains of the phospholipids and maintain the head group spacing. As a result, sugars diminish the interactions between water and phospholipids and then replace the water [118,119]. 

Under anhydrous conditions, sugars are a good replacement for water. Multiple hydrogen bonds are formed between the sugars and lipids at the surface of the lipid bilayer without altering the lipid bilayer structure. Sugars can interact with different lipids simultaneously, interacting with phospholipid polar groups (P=O and/or C=O) and methyl groups of the lipid choline moiety. In vitrification, sugar solutions become freeze-concentrated during freezing, forming a stable glass matrix upon removal of water, resulting in a freeze-dried cake trapped in the glass matrix of sugar [120]. The glass matrix with low mobility and high viscosity protects lipid bilayers from ice crystal-induced damage. Furthermore, the sugar glass matrix inhibits the lipid phase transition-mediated conformational changes [121]. Osmotic and volumetric effects are two key properties during vitrification that reduce the mechanical stress by preventing the adjacent bilayer’s close contact when the lipid membrane aggregates in proximity [122]. The advances in lyophilization techniques, including manometric temperature measurement using SMART freeze-drying and process analytical techniques for monitoring critical process parameters, help in meeting the better storage requirements for mRNA-LNP-based vaccines.

### 6.2. Polymer Nanocarriers

Typically, similar to lipid-based carriers, polymer carriers for mRNA delivery utilize electrostatic attraction forces (between the positive charge of polymer and negative charge of mRNA) for self-assembly of mRNA polyplexes. mRNA polyplexes are attractive in the exploration of mucosal vaccination owing to the recovery of their structure after aerosolization. Compared to lipid-based systems, mRNA polyplexes form more rigid supramolecular structures and have high molecular weight and slower polymeric chain mobility, which provides superior stability [123].

Palamà et al. [124] developed poly(ε-caprolactone) nanoparticles by the emulsion–diffusion–evaporation method for intracellular delivery of GFP (Green Fluorescent Protein)-mRNA. The protamine-mRNA complex was formed prior to the particle assembly for better stability, controlled release, and higher loading of mRNA. The nanoparticles with a core shell structure have an inner core of mRNA covered by a poly(ε-caprolactone) layer that offers greater stability and a stealth property. The authors stated that poly(ε-caprolactone) nanoparticles have the potential to address mRNA instability issues. Polymer carriers for mRNA delivery have been struggling with cytotoxicity, partially due to polymer cationic charge. Modification of cationic-charge polymers with polyethylene glycol chains can improve the delivery of cargo in vitro and in vivo and alleviate cytotoxicity [25]. Furthermore, the innate heterogeneity of polymeric carriers and their relatively low gene transfer efficiency limit clinical translation and large-scale production of polymeric mRNA vaccines [125]. Scaffold-based mRNA vaccine delivery has been exploited owing to patient compliance and less invasive vaccination. Yan et al. [126] reported an injectable chitosan alginate gel scaffold for mRNA vaccine delivery. Lipoplex complexes are formed by the complexation of single-stranded mRNA with nanoparticles of liposomal carrier. The mRNA lipoplexes are then loaded onto lyophilized chitosan–alginate scaffolds followed by a rehydration step. Furthermore, the mRNA release kinetics from the gel and immunization efficiency of gel-mRNA were determined. The results suggest that mRNA vaccine delivery by a scaffold-based method could be a potential alternative to traditional immunization methods.

Poly(ethyleneimine) has been widely used for mRNA vaccine delivery. The optimization of the poly(ethyleneimine) structure provides high gene transfection efficiency. Poly(ethylene imine) properties, such as buffer capacity over a wide pH range and a higher protonation ratio of amino groups at low pH, aid in nucleic acid complexation [127]. Despite the excellent efficacy, the application of poly(ethyleneimine) is limited by its toxicity and the interaction behavior with negatively charged serum proteins, which causes protein aggregation. Incorporation of PEG into the formulation, use of low-molecular-weight polymer form (polyethyleneimine of approx. 2kDa), conjugation to cyclodextrin, and disulfide linkage are some strategies which can mitigate polyethyleneimine toxicity [35].

The low-molecular-weight polyethyleneimine (2k) has been used for delivery of HIV-gag mRNA to dendritic cells and BALB/c mice. Following subcutaneous injection in vivo, the formed mRNA-low MW polyethyleneimine complexes have the potential to induce antigen-specific immune responses [35]. The intranasal administration of an mRNA vaccine based on 2 kDa polyethyleneimine successfully delivered an mRNA-encoding HIV gp120 antigen and induced a systemic immune response [128]. Modifications of polyethyleneimine chemical structure by complexing with several cyclodextrins have been investigated to determine the effect of polyethyleneimine’s chemical structure on the delivery of mRNA to target sites (lymph nodes) and consequent immune responses [129]. Tan et al. [129] synthesized a β-cyclodextrin (β-CD) and branched polyethyleneimine (2 kDa) conjugate for the delivery of an mRNA vaccine. The formed complex efficiently encapsulates mRNA and provides high transfection efficiency by passing through the plasma membranes and escaping from the endosomes.

Other polymeric carriers, such as poly(-amino ester), chitosan, polyamidoamine, and poly(2-propyl acrylic acid), have been investigated in addition to polyethyleneimine. Because of the high amine density on their periphery, poly(-amino ester)s, a biodegradable polymer, efficiently forms mRNA complexes by forming hyper-branched tree-like spherical dendrimers [130]. Chitosan (CS)-based nanoparticles for delivery of nucleic acids (mRNA) have been studied previously. However, their limited ability to escape endosomes hinders the delivery of mRNA. Chitosan is a biocompatible cationic biopolymer derived from chitin that interacts electrostatically with nucleic acids and can be amenable to chemical modifications. The charge density or degree of deacetylation, molecular weight, and the amine-to-phosphate ratio (N:P) are chitosan parameters, which can affect the transfection efficiency of SiRNA-CS-based systems [131]. The bioactivity of CS-mRNA nanoparticles is improved by 4–10× after coating with sulfated hyaluronic acid and adding trehalose [132]. Though the in vitro and in vivo delivery of chitosan nanoparticles are efficient, their colloidal stability and endosomal escape potential are less compared to the lipid nanoparticles for the delivery of mRNA [132]. Polyamidoamine (PAMAM) dendrimers are highly branched (methyl acrylate and ethylenediamine, and end with amine and carboxyl terminal groups) cationic polymers and are biocompatible, allowing for entrapment with nucleic acids. Chahal et al. [133] developed in vitro transcribed conventional unmodified mRNA and modified PAMAM dendrimer nanoparticle (MDNP) for RNA vaccine delivery. The modified dendrimer nanoparticles can provide protective immunity against a broad spectrum of lethal pathogens, including the H1N1 influenza virus and the Ebola virus. The results showed that modified dendrimer nanoparticles induced protective immune responses by providing multiple antigens in mice over a range of disease models.

Poly(ε-caprolactone) is one attractive polymer for mRNA application, approved by the Food and Drug Administration (FDA). The nanoparticles composed of poly(ε-caprolactone) possess low in vitro and in vivo toxicity, high colloidal stability in the biological fluids, controlled release behavior of the encapsulated cargo, and excellent cellular uptake via endocytosis [134]. Biodegradable polymers such as polyglucin, a glucose polymer, and spermidine, a polyamine that occurs in all living organisms, have been adopted for delivery of mRNA vaccines. A polyglucin:spermidine conjugate at a charge ratio of 5:1 self-assembled with an mRNA-encoding SARS-CoV-2 RBD antigen protected the entrapped mRNA from degradation by nuclease, allowed it to be stored at 4C in lyophilized form without loss of nucleic acid activity, which is an important consideration for vaccine storage and transportation [135]. 

E. Jeandupeux et al. [136] investigated the incorporation of poly(2-propyl acrylic acid) PPAA, an anionic polymer with membrane lytic properties at acidic pH, into chitosan mRNA nanoparticles to improve the bioactivity and facilitate endosomal escape. The ternary (CS/mRNA/PPAA) nanoparticles were evaluated for particle size, polydispersity index, z-potential, and in vitro transfection efficiency. The ternary nanoparticles showed an expression level of 86% at pH 6.5 without showing any metabolic toxicity compared to the lipid control (LipofectamineTM MessengerMaxTM (LP-MM) mRNA lipid nanoparticles), for which the bioactivity is only 75%. It was hypothesized that PPAA enhances bioactivity by either increasing endosomal release or by decreasing the CS/mRNA complex stability. While a growing amount of research suggests the use of various polymeric vectors for mRNA delivery, it should be noted that there are currently no comprehensive comparative studies that can direct the ideal formulation for effective delivery of an mRNA vaccine using polymeric carriers.

### 6.3. Incorporation of Adjuvants to Lipid Nanoparticles 

Adjuvants are incorporated into vaccines in order to enhance the immune response through the activation of cell-specific receptors that, in turn, facilitate antigen presentation. Insoluble aluminum salts have been traditionally used as adjuvants. However, they are associated with numerous limitations, such as ineffectiveness towards certain antigens and inability to initiate potent cellular immune responses that necessitates the need of alternatives [137]. The toll-like receptors (TLRs) that are expressed on antigen-presenting cells serve as the major target for adjuvant development due to their ability to enhance cytokine production following their activation. This, in turn, triggers the immune system, which leads to the enhancement of the potency of vaccines [138].

Peptides have been incorporated as adjuvants into lipid nanoparticles as well. Xiang et al. synthesized lymph node-targeted melittin–lipid nanoparticles that were capable of stimulating the abundant antigen-presenting cells located in the lymph nodal region, thus improving cancer immunotherapy outcomes. In comparison to free-melittin, α-melittin–lipid nanoparticles exhibited a 3.6-fold increase in the stimulation of CD8+ T-helper cells that can exert effective action against tumor cells. The good stability and lack of side effects of α-melittin make it an ideal lymph node-targeted nanovaccine with translational potential that can induce a systemic anti-tumor response [139]. In some studies, monophosphoryl lipids derived from bacteria have been incorporated as adjuvants. Ravindran et al. formulated a liposomal preparation by incorporation of a soluble leishmanial antigen (SLA) with an adjuvant monophosphoryl lipid–trehalose dicorynomycolate, that is capable of potentiating an immune response against visceral leishmaniasis. The adjuvanted liposomal formulation demonstrated a significantly greater level of protection against Leishmania donovani in the liver and spleen of BALB/c mice. Cellular immune responses including the induction of IFN-γ and Ig2a antibodies were evident even after four months of vaccination that serves as potential evidence for the ability of the developed formulation to confer long-term immunity [140]. Chikh et al. explored synthetic methylated cytosine–guanine motifs containing oligonucleotides for the potential use as vaccine adjuvants and concluded that encapsulation of the adjuvant within stabilized lipid nanoparticles and enhanced its immunostimulatory activity. These adjuvants belong to a group of molecules named pathogen-associated molecular patterns (PAMPs) that can be recognized specifically by the pathogen-recognition receptors expressed on antigen-presenting cells. The adjuvants were found to act via a toll-like receptor 9 (TLR-9) signaling pathway that was evident from the preliminary data obtained, which demonstrated an up-regulation of TLR-9 expression, nitric oxide induction, and endosomal maturation [141]. In a study conducted by Lee et al., an adjuvant known as Pam-3 was incorporated into lipid nanoparticles to facilitate mRNA-mediated cancer immunotherapy. Ionizable lipids have emerged as promising carriers for mRNA delivery. These lipids possess a neutral charge under physiological conditions and a positive charge under acidic conditions, which enables easy incorporation of the mRNAs at a low pH to produce lipid nanoparticles with a characteristically high encapsulation efficiency. The formulated nanoparticles exhibited successful expression of tumor antigens, ultimately leading to the stimulation of immune responses. Thus, the Pam-3-incorporated lipid nanoparticles provided a synergistic effect for the prevention of tumors by mRNA vaccines [142]. In some cases, based on the nature of lipids that have been incorporated, the lipid nanoparticles themselves act as adjuvants, resulting in activation of the immune system. Mohamed et al. demonstrated an enhancement in the efficacy of mRNA and protein subunit vaccines upon encapsulation in lipid nanoparticles, owing to the induction of T-follicular helper cells and activation of humoral responses. Incorporation of an ionizable lipid component was found to be critical in eliciting the immune responses. The results obtained from comparative studies demonstrated the superiority of the formulated lipid nanoparticles over currently approved adjuvants such as MF59 [143].

The charge of a biomaterial has a significant effect on its interaction with the immune system. The ability of cationic lipids such as 1,2-dioleoyl-3-trimethylammonium propane to stimulate a pro-inflammatory response was demonstrated by Kedmi et al. The induction of Th1 cytokines including IL-2, IFN-γ, and TNF-α by the cationic nanoparticles was found to be 10- to 75-fold higher than control. The positive charge of the lipids enables binding and condensation of nucleic acids through electrostatic interactions, thus facilitating the delivery of the payload across the cellular membrane into the cytoplasm. Thus, cationic lipids are one of the most widely studied non-viral vectors intended for the delivery of nucleic acids, mRNAs, and small interfering RNAs [143]. Zhang et al. have developed a nanovaccine with C1 lipid nanoparticles that possesses a self-adjuvant feature, for the delivery of an mRNA vaccine with anti-tumor efficacy. The co-delivery of the antigen and the immune-potentiating adjuvant promotes the uptake by antigen-presenting cells that, in turn, activates the TLR4 signaling. Further, the in vivo biodistribution studies demonstrated that strong localization of the nanovaccine occurred in the lymph nodes and lungs. Estimation of the in vivo efficacy represented the highest concentration of CD8+ T cells in the lymph nodes, thus confirming the potentiation of immune response after administration of the nanovaccine [144]. The formulation of nucleoside-modified mRNAs in lipid nanoparticles have proved to be an efficacious mode of immunization against infectious diseases. It has been established that, in the immunization against AIDS, mRNA-lipid nanoparticles elicit either the same or enhanced magnitude of immune response, i.e., high titers of serum HIV-1-binding antibodies in comparison to recombinant-protein vaccines. The induction of antibodies against HIV was found to be persistent for a minimum of 41 weeks. Thus, adjuvant-incorporated lipid nanoparticles hold a promising potential in the production of single or multi-component vaccines against various infections [145].

### 6.4. Antigen-Presenting Cells Targeting

Lipid nanoparticles are emerging as tremendously effective carrier systems for the delivery of vaccines owing to their versatile characteristics such as biocompatibility, high loading efficiency, and tailorable surface properties. The successful development of two vaccines for combating COVID-19 by Moderna (mRNA-1273) and Pfizer–BioNTech has proved the immense translational value of lipid nanoparticles. Following many years of intensive research, modern lipid nanoparticle technology has emerged to be a clinically advanced system for gene delivery that overcomes the major difficulties associated with conventional gene therapy, including nucleic acid degradation and minimal cellular uptake [146]. Adjuvants are incorporated into vaccines in order to enhance the immune response through the activation of cell-specific receptors that, in turn, facilitate antigen presentation. Insoluble aluminum salts have been traditionally used as adjuvants. However, it is associated with numerous limitations, such as ineffectiveness towards certain antigens and inability to initiate potent cellular immune responses that necessitates the need of alternatives [137]. The toll-like receptors (TLRs) that are expressed on antigen-presenting cells serve as the major target for adjuvant development due to their ability to enhance cytokine production following their activation. This, in turn, triggers the immune system, which leads to the enhancement of the potency of vaccines [138].

Peptides have been incorporated as adjuvants into lipid nanoparticles as well. Xiang et al. synthesized lymph node targeted melittin-lipid nanoparticles that were capable of stimulating the abundant antigen-presenting cells located in the lymph nodal region, thus improving cancer immunotherapy outcomes. In comparison to free-melittin, α-melittin-lipid nanoparticles exhibited a 3.6-fold increase in the stimulation of CD8+ T-helper cells that can exert effective action against tumor cells. The good stability and lack of side effects of α-melittin make it an ideal lymph node targeted nanovaccine with translational potential that can induce a systemic anti-tumor response [139]. In some studies, monophosphoryl lipids derived from bacteria have been incorporated as adjuvants. Ravindran et al. formulated a liposomal preparation by incorporation of a soluble leishmanial antigen (SLA) with an adjuvant monophosphoryl lipid-trehalose dicorynomycolate, that is capable of potentiating an immune response against visceral leishmaniasis. The adjuvanted liposomal formulation demonstrated a significant greater level of protection against Leishmania donovani in the liver and spleen of BALB/c mice. Cellular immune responses including the induction of IFN-γ and Ig2a antibodies were evident even after four months of vaccination that serves as potential evidence for the ability of the developed formulation to confer long-term immunity [140]. Chikh et al. explored synthetic methylated cytosine-guanine motifs containing oligonucleotides for the potential use as vaccine adjuvants and concluded that encapsulation of the adjuvant within stabilized lipid nanoparticles enhanced its immunostimulatory activity. These adjuvants belong to a group of molecules named as pathogen-associated molecular patterns (PAMPs) that can be recognized specifically by the pathogen-recognition receptors expressed on antigen-presenting cells. The adjuvants were found to act via a toll-like receptor 9 (TLR-9) signaling pathway that was evident from the preliminary data obtained, which demonstrated an up-regulation of TLR-9 expression, nitric oxide induction and endosomal maturation [141]. In a study conducted by Lee et al. an adjuvant known as Pam-3 was incorporated into lipid nanoparticles to facilitate mRNA-mediated cancer immunotherapy. Ionizable lipids have emerged as promising carriers for mRNA delivery. These lipids possess a neutral charge under physiological conditions and a positive charge under acidic conditions, which enables easy incorporation of the mRNAs at a low pH to produce lipid nanoparticles with a characteristically high encapsulation efficiency. The formulated nanoparticles exhibited successful expression of tumor antigens, ultimately leading to the stimulation of immune responses. Thus, the Pam-3 incorporated lipid nanoparticles provided a synergistic effect for prevention of tumor by mRNA vaccines [142]. In some cases, based on the nature of lipids that have been incorporated, the lipid nanoparticles themselves act as adjuvants, resulting in activation of the immune system. Mohamed et al. demonstrated an enhancement in the efficacy of mRNA and protein subunit vaccines, upon encapsulation in lipid nanoparticles owing to the induction of T-follicular helper cells and activation of humoral responses. Incorporation of an ionizable lipid component was found to be critical in eliciting the immune responses. The results obtained from comparative studies demonstrated the superiority of the formulated lipid nanoparticles over currently approved adjuvants such as MF59 [143].

The charge of a biomaterial has a significant effect on its interaction with the immune system. The ability of cationic lipids such as 1,2-dioleoyl-3-trimethylammonium propane to stimulate a pro-inflammatory response was demonstrated by Kedmi et al. The induction of Th1 cytokines including IL-2, IFN-γ and TNF-α by the cationic nanoparticles was found to be 10 to 75-fold higher than control. The positive charge of the lipids enables binding and condensation of nucleic acids through electrostatic interactions, thus facilitating the delivery of payload across the cellular membrane into the cytoplasm. Thus, cationic lipids are one of the most widely studied non-viral vectors intended for the delivery of nucleic acids, mRNAs and small interfering RNAs [143]. Zhang et al. have developed a nanovaccine with C1 lipid nanoparticles that possesses a self-adjuvant feature, for the delivery of an mRNA vaccine with anti-tumor efficacy. The co-delivery of antigen and the immune-potentiating adjuvant promotes the uptake by antigen-presenting cells that, in turn, activates the TLR4-signalling. Further, the in vivo biodistribution studies demonstrated that strong localization of the nanovaccine occurred in the lymph nodes and lungs. Estimation of the in vivo efficacy represented the highest concentration of CD8+ T cells in the lymph nodes, thus confirming the potentiation of immune response after administration of the nanovaccine [144]. The formulation of nucleoside-modified mRNAs in lipid nanoparticles have proved to be an efficacious mode of immunization against infectious diseases. It has been established that, in the immunization against AIDS, mRNA-lipid nanoparticles elicit either the same or enhanced magnitude of immune response, i.e., high titers of serum HIV-1 binding antibodies in comparison to recombinant-protein vaccines. The induction of antibodies against HIV was found to be persistent for at least 41 weeks. Thus, adjuvant incorporated lipid nanoparticles hold a promising potential in the production of single or multi-component vaccines against various infections [145].

### 6.5. Self-Amplifying mRNA Vaccines

Lipid nanoparticles are emerging as tremendously effective carrier systems for the delivery of vaccines owing to its versatile characteristics such as biocompatibility, high loading efficiency and tailorable surface properties. The successful development of two vaccines for combating COVID-19 by Moderna (mRNA-1273) and Pfizer-BioNTech has proved the immense translational value of lipid nanoparticles. Following many years of intensive research, modern lipid nanoparticle technology has emerged to be a clinically advanced system for gene delivery that overcomes the major difficulties associated with conventional gene therapy, including nucleic acid degradation and minimal cellular uptake [146]. Adjuvants are incorporated into vaccines in order to enhance the immune response through the activation of cell-specific receptors that, in turn, facilitate antigen presentation. Insoluble aluminum salts have been traditionally used as adjuvants. However, it is associated with numerous limitations, such as ineffectiveness towards certain antigens and inability to initiate potent cellular immune responses that necessitates the need of alternatives [137]. The toll-like receptors (TLRs) that are expressed on antigen-presenting cells serve as the major target for adjuvant development due to their ability to enhance cytokine production following their activation. This, in turn, triggers the immune system, which leads to the enhancement of the potency of vaccines [138].

The charge of a biomaterial has a significant effect on its interaction with the immune system. The ability of cationic lipids such as 1,2-dioleoyl-3-trimethylammonium propane to stimulate a pro-inflammatory response was demonstrated by Kedmi et al. The induction of Th1 cytokines including IL-2, IFN-γ and TNF-α by the cationic nanoparticles was found to be 10 to 75-fold higher than control. The positive charge of the lipids enables binding and condensation of nucleic acids through electrostatic interactions, thus facilitating the delivery of payload across the cellular membrane into the cytoplasm. Thus, cationic lipids are one of the most widely studied non-viral vectors intended for the delivery of nucleic acids, mRNAs and small interfering RNAs [142]. Zhang et al. have developed a nanovaccine with C1 lipid nanoparticles that possesses a self-adjuvant feature, for the delivery of an mRNA vaccine with anti-tumor efficacy. The co-delivery of antigen and the immune-potentiating adjuvant promotes the uptake by antigen-presenting cells that, in turn, activates the TLR4-signalling. Further, the in vivo biodistribution studies demonstrated that strong localization of the nanovaccine occurred in the lymph nodes and lungs. Estimation of the in vivo efficacy represented the highest concentration of CD8+ T cells in the lymph nodes, thus confirming the potentiation of immune response after administration of the nanovaccine [144]. The formulation of nucleoside-modified mRNAs in lipid nanoparticles have proved to be an efficacious mode of immunization against infectious diseases. It has been established that, in the immunization against AIDS, mRNA-lipid nanoparticles elicit either the same or enhanced magnitude of immune response, i.e., high titers of serum HIV-1 binding antibodies in comparison to recombinant-protein vaccines. The induction of antibodies against HIV was found to be persistent for at least 41 weeks. Thus, adjuvant incorporated lipid nanoparticles hold a promising potential in the production of single or multi-component vaccines against various infections [145].

## 7. Shortcomings of mRNA Vaccines 

### 7.1. Duration of Antibody Response

Antigens produced after mRNA vaccination are taken up by APCs and transported to lymph nodes. Here, interactions between B cells, APCs, and follicular helper T cells (TFH cells) encourage the formation of a germinal center. The B cells then proliferate in the germinal center and differentiate and mutate to produce high-affinity neutralizing antibodies against the pathogen. This cascade of biochemical immunological reactions is crucial for a durable antibody, which corresponds to a long-term duration of action against the infectious disease [147]. Several promising mRNA vaccines are in development which have promising strategies that actively target APCs. Targeting LNPs with APC cell-specific ligands, mAbs, and peptides are some of the strategies being explored to increase the immune response generated by mRNA vaccines [148,149]. Additionally, altering mRNA vaccine pharmacokinetic properties by prolonging the translation of antigenic mRNA has now emerged as a promising tool to enhance the antibody response [150]. mRNA vaccines have elicited potent germinal center immunogenic reactions and TFH cell induction in preclinical studies against HIV-1, SARS-CoV-2, Zika virus, and influenza virus [151,152,153,154]. Although these results are promising, the duration of antibody response is a complex phenomenon which will vary highly from antigen to antigen. Additionally, evaluating the duration of immune response by MRNA vaccines requires longer-term data for a comprehensive understanding.

### 7.2. Safety

Overall, the current mRNA vaccines have promising safety profiles as demonstrated in clinical trials and post-approval real population data. These vaccines have only mild or moderate adverse events as seen in clinical trials. However, there have been some scattered safety incidents that require further optimization of mRNA vaccines and all its components. For instance, CureVac’s protamine-based rabies vaccine, CV7201, caused adverse effects in 78% of participants [29]. This resulted in CureVac adopting LNPs as their primary and preferred delivery vehicle for their next rabies candidate, CV7202 [155]. As with most medications, the adverse reactions to mRNA vaccines have often increased and escalated with dose. For example, in Phase I trials of Moderna’s influenza H10N8 vaccine, adverse events were observed from the 400 μg. Hence, they continued with a lower dose of up to 100 μg [156]. In Phase I trials of CV7202, a 5 μg dose had a high reactogenicity; hence, 1 μg was the highest dose administered to the subjects. 

Mild anaphylactic reactions have been seen in 4.7 per million COVID-19 vaccinations, with 2.5 per million vaccinations with the Moderna vaccine and 2.2 per million with Pfizer–BioNTech vaccine [157]. These are significantly higher than what is typically seen with traditional vaccines [158]. Scientists have proposed that this allergic response can be attributed to pre-existing antibodies that the patients have against the PEGylated lipids which are used in LNPs. These antibodies can be formed in the body in response to the presence of PEG in many consumer products, such as toothpastes and shampoos. Although PEG is safe, it is rumored to activate humoral immunity in a subset of the population in a T cell-independent manner. It does this by directly crosslinking the B cell receptor and introducing IgM production [159]. Anti-PEG antibodies are reported in 40% of the population, which can accelerate and heighten the risk of allergic reactions and impede vaccine efficacy [160]. The CDC recommends that mRNA vaccines should not be given to people with a history of allergic response to the Pfizer–BioNTech or Moderna vaccines. Since some components of mRNA vaccine formulations can cause allergic reactions in a fraction of the population, the formulation components should be re-engineered for enhanced safety profiles.

### 7.3. Maternal/Neonatal Vaccination

The immune system during pregnancy and in the neonatal stage of infants is highly dynamic and evolving, which can increase a person’s predisposition to infectious diseases. Zika virus can infect cortical neurons and glial cells in the developing fetus, resulting in cell death, neuroinflammation, and severe congenital malformations of the fetus [161]. Cytomegalovirus infection can causes complications in approximately 1% of pregnancies leading to congenital disabilities, in addition to neurological impairment in infants [161]. Extremely rare in utero transmission of the SARS-CoV-2 virus has also been reported. Its implications on maternal and neonatal health is under investigation [162,163]. To address these shortcomings, maternal vaccination has emerged as a tool to advance maternal health and reduce the burden of neonatal morbidity. Maternal IgG antibodies can readily cross the placental barrier by binding to the neonatal crystallizable fragment (Fc) receptor and enter fetal circulation. This protects the fetus from pathogens and other infectious diseases. Several preclinical studies have demonstrated that maternal vaccination with mRNA-LNPs prevented Zika virus transmission to the fetus in pregnant mice, group A, and group B streptococci, and protected mouse neonates from herpesvirus [164,165,166,167].

### 7.4. Geriatric Vaccinations

It is expected that by 2050, the proportion of the world population over 60 years is expected to double from 12% to 22%. Vaccines for this population are much required, as many infectious diseases affect the elderly disproportionately. For example, 70–90% of influenza-related mortalities occurred in people older than 65 years, and COVID-19 is significantly (65 times) more fatal in patients older than 65 years than it is in younger patients [168,169]. Geriatric populations are more difficult to immunize via vaccinations, since the patients’ age adversely affects the innate and adaptive immune system [170]. The adaptive immune responses that arise after an infection are often inadequate due to impaired cytokine signaling, in addition to impaired physiological and cellular changes. These changes can include fewer naive B and T cells, higher susceptibility to T cell apoptosis, diminished T cell receptor diversity, and reduced expression of crucial receptors such as CD28 on cytotoxic CD8+ T cells [171,172,173]. 

mRNA vaccines might be just the solution for boosting geriatric immunity. These vaccines might offer robust efficacy to all age groups, especially the elderly, as seen in the Phase III trial of the Pfizer–BioNTech vaccine candidate BNT162b2. The vaccine elicited more than 93% efficacy across all treatment groups well-defined by age [174]. Similarly, the Moderna vaccine mRNA-1273 was also extremely effective, and showed 86.4% efficacy in volunteers older than 65 years old, in comparison to 95.6% efficacy in 18–65-year-olds [175]. The design of efficient drug delivery systems is imperative for improving vaccine efficacy in the elderly. mRNA delivery vehicles act as adjuvants and amplify vaccine response by enhancing APC recruitment to the injection site. For example, Novartis’s oil-in-water emulsion MF59, has been utilized as an mRNA delivery vehicle, and can be employed to act as an adjuvant. MF59 amplifies the immune response of influenza vaccines and it has been approved for use in elderly adults [176]. In the elderly population, influenza vaccines adjuvanted with MF59 enhanced the seroconversion and seroprotection rates as compared to non-adjuvanted vaccines [177].

### 7.5. Vaccine Acceptance

Vaccines are effective only if they are administered and if there is a satisfactory vaccine acceptance and a belief in the effectiveness of the vaccines. Nevertheless, public doubts fueled by misinformation threaten the achievement and maintenance of herd immunity and puts the most vulnerable populations, such as the elderly and children, at risk. Declining vaccination coverage and administrations can lead to the re-emergence of life-threatening diseases that are otherwise now extinct. For example, measles, which has been completely eradicated from the USA in 2000, has infected more than 1200 people in 2019 due to poor vaccine acquiescence [35]. For COVID-19, due to vast information availability and massive awareness, the vaccine acceptance rates range from 55% to 90% around the world [178]. The current acceptance rates in the USA are 56–75%, which may be insufficient to maintain the threshold necessary for herd immunity against SARS-CoV-2 [179,180]. Additionally, in the USA, the mRNA vaccine trials’ high efficacy rates have increased public confidence in mRNA vaccines. 

### 7.6. Access to Vaccines

Affordable and easy access to vaccines is the greatest challenge in achieving prevalent protection against infectious diseases, specifically in low-income countries. This access is further restricted due to the cold-storage requirements of the SARS-CoV-2 mRNA vaccines. During the lethal 2014–2016 Ebola virus outbreak in West Africa, vaccines requiring −80 °C storage were supplied in the Democratic Republic of the Congo by means of portable and reusable Arktek freezers, permitting the vaccine to be administered to 400,000 people. Such cold-storage technologies are capable for the rapid deployment of millions of doses during an epidemic. However, vaccinating billions of people amidst a continuously evolving pandemic such as COVID-19 requires thermostable vaccines. Two SARS-CoV-2 vaccine candidates have reported to be thermostable in preclinical studies at room temperature [88,181]. If these thermostable mRNA vaccine candidates show promising results in clinical trials, they can immensely simplify global access to mRNA vaccines in the near future. Designing affordable, stable, and effective thermostable mRNA vaccines is the need of the hour.

## 8. Conclusions

Decades of development and research in mRNA design and its delivery technology have made mRNA vaccines an astonishing tool for combating pandemics and existing infectious diseases. The first two mRNA vaccines to combat SARS-CoV-2 were developed at an unexpected rate. These vaccines have exceeded expectations and laid a strong foundation and essential groundwork for the future of mRNA vaccines. It is evident from the plethora of clinical trials for mRNA vaccines that these can be head-to-head or even replace the conventional vaccine platform in the near future. mRNA technology has the potential for the development of more effective vaccines against persistent and challenging pathogens and treat various cancers in the near future. Nevertheless, advancement in mRNA delivery technologies will be required for more effective, safer, and cold-chain-free mRNA vaccines, having the capacity to vaccinate billions of populations across boundaries. Further research on how the mRNA vaccines impact innate immune responses needs to be investigated. The abundance of positive safety and efficacy data for the approved mRNA vaccines, together with a proven path for regulatory approval, lights a hope within the scientific community that mRNA therapeutics indeed have an immense potential to transform modern biotherapeutic approaches to vaccination, protein replacement therapy, and cancer immunotherapy [35].

## Figures and Tables

**Figure 1 ijms-24-02700-f001:**
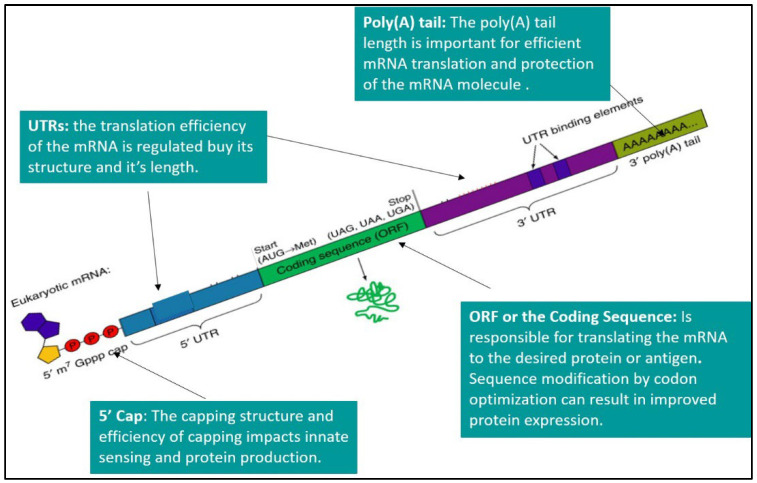
mRNA molecule structural components [25].

**Figure 2 ijms-24-02700-f002:**
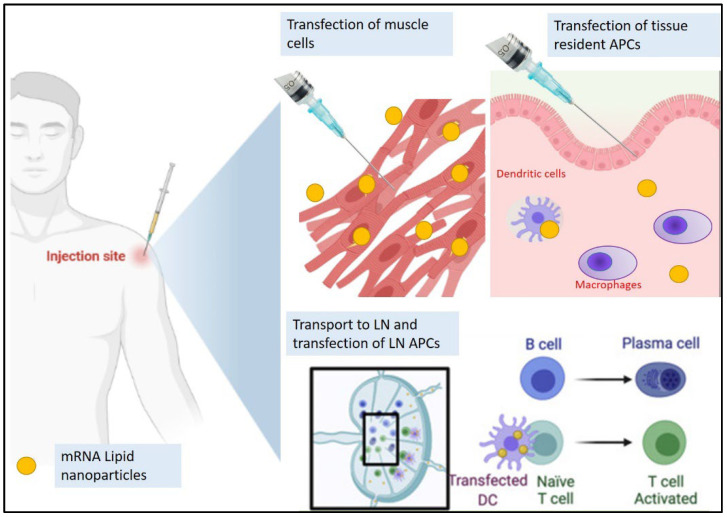
mRNA lipid nanoparticles’ (mRNA-LNPs) site of intramuscular administration and modes of action of the mRNA-LNPs. mRNA-LNP vaccines can transfect muscle cells and transfect the tissue-resident antigen-presenting cells (APCs) near the injection site. Additionally, mRNA-LNP vaccines can flow into lymph nodes (LNs) and transfect the LN-resident cells, resulting in activation of T and B cells. Adapted with permission from [25].

**Figure 3 ijms-24-02700-f003:**
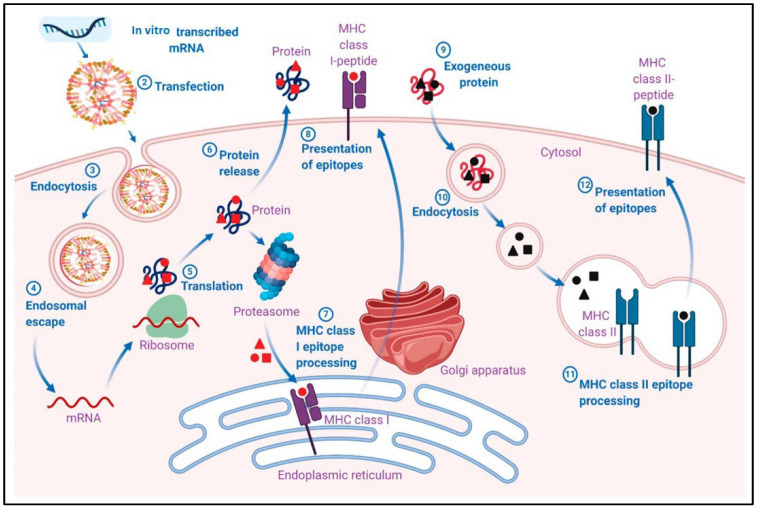
Pharmacological mechanism of adaptive immune responses induced by mRNA-LNP vaccines. (1) In vitro transcribed mRNA is encapsulated into a lipid nanoparticle (LNP). (2) Transfection of mRNA-LNP vaccine molecules into the host cells, using specialized lipids on the surface of the LNPs. (3) Endocytosis of mRNA-LNP. (4) Endosomal escape of mRNA to the cytosol after endocytosis-mediated internalization. (5) Translation of the mRNA by the host cell ribosomes into the desired antigen protein intracellularly. (6) Antigenic protein released outside the cell, or the antigenic protein is degraded by a proteosome, exposing the antigenic sites. (7) Major histocompatibility complex I (MHC I) epitope presentation of the MHC I to the cell membrane for antigen presentation (APC). MHC I presents the epitope to CD8+ T cells. (9) The exogenous protein released earlier can get degraded and presented via MHC II epitopes. The extracellular antigen can get recognized by B cells, leading to B cell maturation [25].

**Figure 4 ijms-24-02700-f004:**
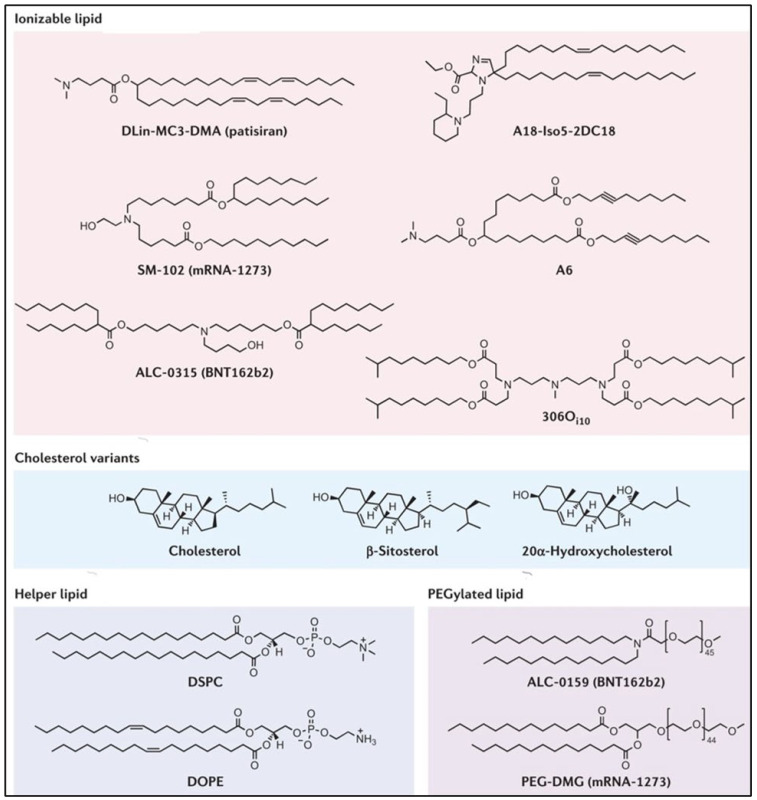
Components of lipid nanoparticles including ionizable lipids, cholesterol, helper lipids, and PEGylated lipids [35].

**Figure 5 ijms-24-02700-f005:**
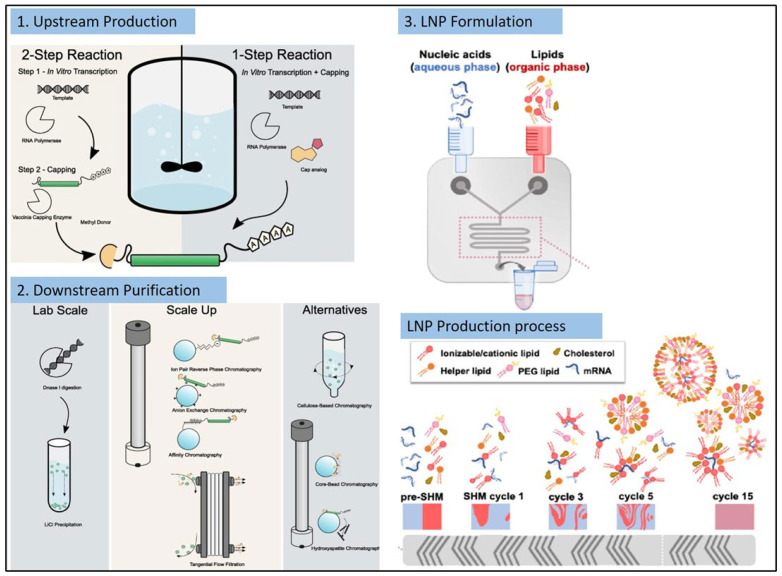
The steps and stages of an mRNA vaccine manufacturing process. mRNA vaccine production can be divided into three phases: upstream mRNA manufacturing, downstream mRNA purification, and formulation of mRNA lipid nanoparticles. mRNA production can be performed in a one-step co-transcriptional reaction, where a capping reagent is used, or in a two-step reaction, where the enzymatic capping is performed. mRNA purification process at a smaller lab scale consists of DNase I digestion enzyme followed by LiCl precipitation of the mRNA. Purification of mRNA at a large scale involves utilizing well-established chromatographic methods coupled with tangential flow filtration (TFF). Finally, the formulation of mRNA vaccines consists of mixing mRNA aqueous solution with lipid solution in a non-aqueous phase. This causes self-assembly of the lipid nanoparticles (LNPs) and encapsulates the negatively charged mRNA within the core of the LNPs. The mixing of the mRNA and the lipid molecules in a staggered herringbone micromixer (SHM) occurs in various cycles which results in the formation of the final mRNA-LNP vaccines. Adapted with permission from [86,90].

**Table 1 ijms-24-02700-t001:** Ongoing Clinical Trials With mRNA Vaccines (Excluding COVID-19 Vaccines).

Vaccine	Formulation Type/Route of Administration	Indication	Clinical Trial Number	Phase	Sponsor	Status
eOD-GT8 60mer mRNA	Nanoparticle/Intraperitoneal	HIV	NCT05414786	1	International AIDS Vaccine Initiative	Recruiting
Core-g28v2 60mer mRNA vaccine and eOD-GT8 60mer mRNA vaccine	Nanoparticle/Intramuscular injection	HIV	NCT05001373	1	International AIDS Vaccine Initiative	Recruiting
BG505 MD39.3 mRNA, BG505 MD39.3 gp151 mRNA, and BG505 MD39.3 gp151 CD4KO mRNA	NA/Intramuscular injection	HIV	NCT05217641	1	National Institute of Allergy and Infectious Diseases (NIAID)	Recruiting
mRNA-1345	Lipid nanoparticle/Intramuscular injection	Respiratory Syncytial Virus (RSV)	NCT05127434	2/3	Moderna	Recruiting
		RSV	NCT04528719	1	Moderna	Recruiting
mRNA-1345 and mRNA-1273.214	Lipid nanoparticle/Intramuscular injection	RSV	NCT05330975	3	Moderna	Recruiting
Influenza vaccines (mRNA-1020, mRNA-1030, and mRNA-1010)	Lipid nanoparticle/Intramuscular injection	Influenza (A and B strains)	NCT05333289	1/2	Moderna	Recruiting
		Influenza (A and B strains)	NCT05375838	1/2	Moderna	Recruiting
mRNA-1010	Lipid nanoparticle/Intramuscular injection	Seasonal influenza	NCT04956575	1/2	Moderna	Recruiting
		Seasonal influenza	NCT05415462	3	Moderna	Recruiting
Influenza vaccines {monovalent influenza modRNA vaccine (mIRV), bivalent influenza modRNA vaccine (bIRV AB, bIRV AA, and bIRV BB) quadrivalent influenza modRNA vaccine (qIRV)}	NA/Intramuscular injection	Influenza	NCT05052697	1/2	Pfizer	Recruiting
Seasonal quadrivalent influenza mRNA vaccine CVSQIV	NA/Intramuscular injection	Influenza	NCT05252338	1	CureVac AG	Recruiting
Self-amplifying ribonucleic acid (saRNA) vaccines (PF-07852352, PF-07836391, PF-07836394, PF-07836395, PF-07836396, and PF-07867246)	NA/Intramuscular injection	Influenza	NCT05227001	1	Pfizer	Recruiting
mRNA NA vaccine	NA/Intramuscular injection	Influenza	NCT05426174	1	Sanofi Pasteur	Recruiting
mRNA-1647	NA/Intramuscular injection	Cytomegalovirus infection	NCT05085366	3	Moderna	Recruiting
			NCT04232280	2	Moderna	Recruiting
			NCT05105048	1	Moderna	Recruiting
mRNA -1215	Lipid nanoparticle/Intramuscular injection	Nipah virus	NCT05398796	1	National Institute of Allergy and Infectious Diseases (NIAID)	Recruiting
W_ova1 vaccine	Liposome/Intravenous injection	Ovarian cancer	NCT04163094	1	University Medical Center Groningen	Active, not recruiting
National Cancer Institute (NCI)-4650	Lipid nanoparticle/Intramuscular injection	Cancer (Melanoma, Colon, Gastrointestinal, Genitourinary, and Hepatocellular)	NCT03480152	1/2	National Cancer Institute (NCI)	Terminated
BNT113	Liposome/Intradermal vaccine	Carcinoma, Squamous Cell, Head and Neck Neoplasm, Cervical Neoplasm, Penile Neoplasms Malignant	NCT03418480	1/2	University of Southampton	Recruiting
	Liposome/Intradermal vaccine	Unresectable Head and Neck Squamous Cell Carcinoma Metastatic Head and Neck Cancer Recurrent Head and Neck Cancer	NCT04534205	2	BioNTech SE	Recruiting
BNT111	NA/Intravenous infusion	Melanoma Stage III Melanoma Stage IV Unresectable Melanoma	NCT04526899	2	BioNTech SE	Recruiting
Individualized Cancer RNA Immunotherapy (IVAC^®^) vaccines: IVAC_W_bre1_uID and IVAC_W_bre1_uID/IVAC_M_uID	NA/Intravenous injection	Triple Negative Breast Cancer (TNBC)	NCT02316457	1	BioNTech SE	Active, not recruiting
RNA tumor vaccine	NA/Intramuscular injection	Solid tumor	NCT05202561	1	First Affiliated Hospital Bengbu Medical College	Recruiting
mRNA-1893	Solution/Intramuscular injection	Zika virus	NCT04917861	1	Moderna	Recruiting
DC-006 vaccine loaded with amplified cancer stem cell mRNA	NA/Intranodal injection	Recurrent Epithelial Ovarian Cancer	NCT01334047	1/2	Steinar Aamdal	Terminated
Lipo-MERIT	NA/IV injection	Melanoma	NCT02410733	1	BioNTech SE	Active, not recruiting
mRNA-4157	Lipid nanoparticles/intramuscular injection	Melanoma	NCT03897881	2	Moderna	Active, not recruiting
mRNA-1653	NA/Intramuscular injection	Human Metapneumovirus and Human Parainfluenza Infection	NCT04144348	1	Moderna	Recruiting
mRNA-1189	Lipid nanoparticles/intramuscular injection	Epstein-Barr Virus Infection	NCT05164094	1	Moderna	Recruiting
Dendritic cells loaded with mRNA	Dendritic cell vaccine/NA	Prostate cancer	NCT01197625	1/2	Oslo University Hospital	Active, not recruiting
Langerhans-type dendritic cells with mRNA	Dendritic cell vaccine/Intradermal injection	Melanoma	NCT01456104	1	Memorial Sloan Kettering Cancer Center	Active, not recruiting
RNA-lipid particle (RNA-LP) vaccines	Liposome/Intravenous infusion	Adult Glioblastoma	NCT04573140	1	University of Florida	Recruiting
Autologous dendritic cells electroporated with WT1 mRNA	Dendritic cell vaccine/Intradermal injection	Acute Myeloid Leukemia	NCT01686334	2	Zwi Berneman	Recruiting
		Myelodysplastic Syndromes Acute Myeloid Leukemia	NCT03083054	1/2	University of Campinas, Brazil	Active, not recruiting
WT1 mRNA-loaded autologous monocyte-derived dendritic cells	Dendritic cell vaccine/Intradermal injection	High Grade Glioma Diffuse Intrinsic Pontine Glioma	NCT04911621	1/2	University Hospital, Antwerp	Recruiting
Human CMV pp65-LAMP mRNA-pulsed autologous dendritic cells	Dendritic cell vaccine/Intradermal injection	Glioblastoma	NCT03688178	2	Gary Archer Ph.D and Celldex Therapeutics	Recruiting

## Data Availability

Not applicable.

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
