# Peer review of "A Comprehensive Review of mRNA Vaccines"

_ijms, 2023, doi:10.3390/ijms24032700_

Round 1
Reviewer 1 Report
In this review authors have summarized various aspect of mRNA vaccine. They described the structure, pharmacological function of immunity induction, lipid nanoparticles (LNPs), and the upstream, downstream and formulation process of mRNA vaccines manufacturing. The mRNA vaccines in the clinical trials are also described.
The review article has fully and comprehensively addressed the various aspects of this vaccine. However, it is recommended to edit grammatically and apply the following.
It would be helpful to describe how can overcome challenges encountered in the past, such as instability, inefficient delivery, and weak carcinogenicity of mRNA vaccines.
It would be helpful until to date, describe hundreds of cancer vaccines have undergone clinical evaluation and the U.S. FDA has approved three therapeutic cancer vaccines and two prophylactic cancer vaccines in separate table.
Line 237: delete one of two repeated and. “epitope presentation and (8) and translocation”
Lines 248, 249: Please correct the sentence “Methods to deliver mRNA molecules into the cells include techniques like gene guns, electroporation, ex vivo transfection” grammatically.
Lines 327-330: Are these two separate sentences or one sentence? It seems that the first sentence is incomplete.
With regards
Author Response
The authors would like to thank the reviewer for these helpful comments. The careful review by the reviewer has definitely improved the quality of the manuscript
Replies to the comments:
It would be helpful to describe how can overcome challenges encountered in the past, such as instability, inefficient delivery, and weak carcinogenicity of mRNA vaccines.
This is described in two particular segments, which include improvements in the stability and efficiency of the drug substance, the mRNA itself in section 2, and then the further improvements in the final drug product in sections like 3 and 5. Techniques like lyophilization to increase the stability of mRNA LNP's is also discussed. Additionally, the section"Clinical Safety of mRNA-based vaccines" which are the COVID-19 vaccines describes the entire toxicological profile of the approved vaccines in the mRNA category.
It would be helpful to date, to describe hundreds of cancer vaccines have undergone clinical evaluation and the U.S. FDA has approved three therapeutic cancer vaccines and two prophylactic cancer vaccines in a separate table.
This is a very great idea since there is a lot of results to discuss for this topic, and this can form an entirely new review article. For this article, we wanted to focus only on the mRNA vaccines for infectious diseases. mRNA vaccines for cancer fall out of the scope of the current article.
Line 237: delete one of two repeated and. “epitope presentation and (8) and translocation” "translocation word is deleted.
Lines 248, 249: Please correct the sentence “Methods to deliver mRNA molecules into the cells include techniques like gene guns, electroporation, ex vivo transfection” grammatically. Corrected.
Lines 327-330: Are these two separate sentences or one sentence? It seems that the first sentence is incomplete. Corrected
Reviewer 2 Report
This paper reviewed the mRNA structure, its pharmacological function of immunity induction, lipid nanoparticles, mRNA vaccines in the clinical trials, and the upstream, downstream and formulation process of mRNA vaccines manufacturing. The reported topic have relevance in nanomedicine field. This article thoroughly discusses the gap between recent advances in second-generation mRNA vaccines (Lyophilized mRNA lipid nanoparticles, Polymer nanocarriers, Incorporation of adjuvants to lipid nanoparticles, Antigen-presenting cells targeting, Self-amplifying mRNA vaccines) and current clinical trials. In this study, the researchers summarized all the ongoing clinical trials with mRNA based vaccines (excluding, HIV, Influenza, Covid-19 vaccines). The references used in this article are appropriate and relevant. The results, tables and figures are interesting and suitable for publication. The experiments, results and discussion as well as supporting information are well presented.
I believe that this manuscript can be considered for publication in this Journal after minor revisions.
- In the section 5. Second generation mRNA vaccines (Line 705), before the presented classification, please provide a paragraph about the basis of the classification of the second generation vaccines.
Author Response
The authors would like to thank the reviewer for these helpful comments. The careful review by the reviewer has definitely improved the quality of the manuscript
Replies to the comments:
In the section 5. Second generation mRNA vaccines (Line 705), before the presented classification, please provide a paragraph about the basis of the classification of the second generation vaccines.
Lines 707-716 describe this in the revised manuscript. Thank you for this comment. It improves the section surely.